# Layers of immunity: Deconstructing the *Drosophila* effector response

**Faustine Ryckebusch[1], Yao Tian[1], Mylene Rapin[1], Fanny Schüpfer[1], Mark Austin Hanson[1,2]\*, Bruno Lemaitre[1]\***

[1]Global Health Institute, Ecole Polytechnique Fédérale de Lausanne (EPFL), Lausanne, Switzerland; [2]Centre for Ecology and Conservation, University of Exeter, Penryn, United Kingdom

## eLife Assessment

This work provides one of the first **important** attempts to look at *Drosophila* immune responses against bacterial, viral, and fungal pathogens in a way that combines the roles of four major arms in immunity (Imd signaling, Toll signaling, phagocytosis, and melanization) rather than studying them separately. The findings are **compelling** and the tools provided can be used as they are, or built upon, in various contexts.

**\*For correspondence:**
m.hanson@exeter.ac.uk (MAH);
bruno.lemaitre@epfl.ch (BL)

**Abstract** The host innate immune response relies on the cooperation of multiple defense modules. In insects and other arthropods, which have only innate immune mechanisms, four main immune-specific modules have been described in the defense against microbial invaders: the Toll pathway, the Imd pathway, the melanization response, and phagocytosis by plasmatocytes. Our present understanding of their relative importance remains fragmented as their contribution to host defense has never been simultaneously assessed across a large panel of pathogens. Here, we use newly described immune mutants in a controlled genetic background to systematically delete these four immune modules individually, in pairs, or even all four simultaneously. Surprisingly, flies simultaneously deficient in all four immune modules are viable (poor viability), homozygous fertile, and display no overt morphological defects, suggesting these immune mechanisms are not strictly required for organismal development. We assessed the contribution of each module individually and collectively against a diverse panel of viruses, fungi, and bacteria. We find these four modules largely function independently and additively in host defense. We could confirm previous findings on the importance of Imd and Toll, and their antimicrobial peptide and Bomanin (Bom) effectors, against relevant microbes. We also reveal a highly important role of melanization against viruses. Examining microbial load kinetics confirms how these modules contribute to resistance or tolerance against specific microbes. The set of immune-deficient lines provided here offers tools to better assess the role of these immune modules in host defense.

## Introduction

The immune system is made up of a network of tissues and circulating cells whose primary function is to prevent and limit pathogenic infections (*Danilova, 2006*; *Pradeu et al., 2024*; *Schmid-Hempel, 2021*). Although immunity has been studied in many animal species, it has been extensively characterized in only a subset of them such as the nematode *Caenorhabditis elegans*, the fruit fly *Drosophila melanogaster*, zebrafish, rodents, and humans. The traditional understanding of the immune system pictures it as a specialized group of cells and tissues, but this has gradually changed over recent years to a more complex picture of highly redundant multilevel defense modules. We have learned a great

deal in the last few decades on the molecular mechanisms regulating each of these immune modules from recognition and signaling to effector action (e.g. phagocytosis, antimicrobial peptides [AMPs]) (*Pradeu et al., 2024*). However, an immune response encompasses the concomitant activation of several modules, and there has been little investigation of how each module collectively contributes to host defense. Indeed, the immune research field has traditionally focused on one module at a time. This article aims to fill this gap by collectively analyzing the specific contribution of the main immune modules involved in the *Drosophila* systemic response.

The *Drosophila* systemic immune response consists of a set of humoral and cellular reactions in the hemolymph (insect blood) when flies are systemically infected by bacteria, viruses, or fungi (*Liegeois and Ferrandon, 2022*; *Westlake et al., 2024*). A first facet of the systemic immune response is the production of host-defense effectors by the fat body and hemocytes that are secreted into the hemolymph. Among the best-characterized effectors are the AMPs, whose expression is induced to incredibly high levels upon infection in a race to control invading microbes (*Hanson et al., 2019*; *Imler and Bulet, 2005*). AMPs are small cationic peptides that exhibit antimicrobial activity that directly participates in microbial killing. Beyond AMPs, many other proteins and host defense peptides (HDPs) are secreted into the hemolymph, creating a hostile environment for invading pathogens. The systemic antimicrobial response is predominantly regulated at the transcriptional level by the Toll and Imd pathways (*De Gregorio et al., 2002*; *Lemaitre et al., 1996*). In *Drosophila*, the Toll pathway is activated by cell wall components (fungal glucans and Lysine-type peptidoglycan), as well as microbial proteases (*Gottar et al., 2006*; *Issa et al., 2018*; *Leulier et al., 2003*; *Vaz et al., 2019*). Toll signaling culminates in the activation of two NF-κB transcription factors, Dif and Dorsal, which regulate a large set of immune genes (*Ip, 1993*; *Lemaitre et al., 1996*). These encode small peptides such as the antifungal peptide Drosomycin, the Toll-regulated Bomanin peptides, and many other proteins (e.g. serine proteases, serpins, and lipases) (*Clemmons et al., 2015*; *De Gregorio et al., 2002*; *Fehlbaum et al., 1994*; *Ligoxygakis et al., 2003*). Flies lacking the Toll pathway are viable but highly susceptible to infection by Gram-positive bacteria and fungi (*Buchon et al., 2009*; *Lemaitre et al., 1996*; *Rutschmann et al., 2002*). The Imd pathway is activated by DAP-type peptidoglycan produced by Gram-negative bacteria and a subset of Gram-positive bacteria (e.g. *Bacillus*) (*Kaneko et al., 2004*; *Lemaitre et al., 1995*; *Leulier et al., 2003*). The binding of peptidoglycan to receptors of the Peptidoglycan Recognition Protein family (PGRP-SD, PGRP-LC, PGRP-LE) initiates an intracellular signaling cascade, which ultimately activates the NF-κB factor Relish (*Choe et al., 2005*; *Iatsenko et al., 2016*; *Kaneko et al., 2006*; *Westlake et al., 2024*). The Imd pathway regulates the expression of many genes encoding antibacterial peptides and serine proteases (*De Gregorio et al., 2002*; *Lemaitre et al., 1997*). Imd-deficient flies are viable but susceptible to Gram-negative bacterial infection.

In addition to humoral pathways, melanization, and phagocytosis are two complementary mechanisms that contribute to host survival upon systemic infection. Melanization is an arthropod-specific immune response resulting in the rapid deposition of the black pigment melanin at wound or infection sites and the concomitant production of microbicidal reactive oxygen species (*Cerenius et al., 2008*; *Nappi et al., 2009*; *Westlake et al., 2024*). The melanization and Toll pathways are co-activated at the level of the serine proteases Hayan and Persephone, which in turn activate Spatzle-processing enzyme (SPE) of the Toll pathway and serine proteases critical to cleavage of prophenoloxidases (PPOs) (*Dudzic et al., 2019*; *Nakano et al., 2023*; *Shan et al., 2023*), collectively referred to as the Toll-PO SP cascade (*Westlake et al., 2024*). The melanization reaction itself relies on the oxidation of phenols, resulting in the polymerization of melanin. Two PPOs (PPO1 and PPO2) are the primary catalysts for the synthesis of melanin during the melanization response upon systemic infection. Flies deficient for both *PPO1* and *PPO2* are viable but lack hemolymph phenoloxidase activity and exhibit susceptibility to certain Gram-positive bacteria and fungi (*Binggeli et al., 2014*; *Dudzic et al., 2015*). In addition, a third PPO gene (PPO3) is specifically expressed by lamellocytes, specialized hemocytes that differentiate in larvae responding to and enveloping invading parasites (*Dudzic et al., 2015*).

Phagocytosis by both sessile and circulating plasmatocytes (*Drosophila* equivalent of macrophages) is thought to provide a complementary and important host defense (*Melcarne et al., 2019b*; *Ulvila et al., 2011*). The use of hemocyte-deficient flies, via the targeted expression of a pro-apoptotic gene in plasmatocytes only, has shown that hemocytes contribute to survival upon systemic infection to certain bacterial species (*Charroux and Royet, 2009*; *Defaye et al., 2009*; *Stephenson et al., 2022*). Phagocytosis of bacteria greatly relies on two transmembrane receptors, NimC1 and Eater (*Bretscher*

*et al., 2015*; *Kocks et al., 2005*; *Kurucz et al., 2007*; *Melcarne et al., 2019b*). *NimC1, Eater* double mutant larvae are viable, have elevated hemocyte numbers at the larval stage, and their hemocytes can encapsulate and melanize macroparasites; however, their hemocytes are non-sessile and nearly totally phagocytosis deficient. *NimC1, Eater* adult flies have decreased hemocyte numbers and fail to perform phagocytosis, providing a good tool to assess the role of the cellular response (*Melcarne et al., 2019b*; *Melcarne, 2020*).

We have circumstantial evidence that these four immune modules are critical for defense against certain pathogens. However, how, when, and to what extent each of these four immune modules contributes to host defense against specific pathogens is fragmented as they have never been simultaneously assessed. Moreover, the extent to which these findings can be generalized to broad classes of pathogens remains unclear.

In this article, we have used a specific set of mutations principally affecting phagocytosis (*NimC1, Eater*), melanization (*PPO1, PPO2*), the Toll pathway (*spz*), and the Imd pathway (*Rel*) in a controlled genetic background to address their role in host defense. We then recombined these and other mutations to generate double and even quadruple mutants of the four immune modules. This set of tools allowed us to compare the survival of flies deficient in these responses against a broad panel of viruses, fungi, Gram-positive bacteria, and Gram-negative bacteria, to reveal the relative contribution of each immune mechanism to resistance against these pathogens.

## Results

### A set of immune-deficient lines in the iso Drosdel background

Mutations affecting only one of the four main modules of the *Drosophila* systemic immune response have been described. These include *Relish$^{E20}$* (*Rel$^{E20}$*, deficient for the Imd pathway, referred to as *ΔIMD*), *spatzle$^{rm7}$* (*spz$^{rm7}$*, deficient for the Toll pathway, referred to as *ΔTOLL*), *PPO1$^Δ$*, *PPO2$^Δ$* (no melanization, referred to as *ΔMel*), and *NimC1$^1$; eater$^1$* (reduced phagocytosis, referred to as *ΔPhag*). In addition, to blocking phagocytosis, *ΔPhag* flies have a defect in hemocyte sessility, an increased total number of larval hemocytes, and a decreased number of hemocytes at the adult stage (*Bretscher et al., 2015*; *Melcarne et al., 2019b*; *Melcarne, 2020*). We have preferred to use *NimC1$^1$; eater$^1$* to study the contribution of hemocytes rather than other approaches with important indirect effects such as the pre-injection of latex beads to saturate phagocytes (*Elrod-Erickson et al., 2000*), or expression of a pro-apoptotic gene in plasmatocytes (*Charroux and Royet, 2009*; *Defaye et al., 2009*). All the mutations in this study were partly isogenized for seven or more generations into the DrosDel *iso w$^{1118}$* genetic background (*Ferreira et al., 2014*; *Ryder et al., 2004*). We then generated by recombination

**Table 1.** List of mutants used in this study.
'-' indicates the deleted immune module.

| *Drosophila* lines | Reference | Name | Deficient pathway | | | |
| --- | --- | --- | --- | --- | --- | --- |
| | | | IMD | Toll | Phagocytosis | Melanization |
| *iso; iso; Rel$^{E20}$* | *Hedengren et al., 1999* | *ΔIMD* | - | + | + | + |
| *iso; iso; spz$^{rm7}$* | *Lemaitre et al., 1996* | *ΔTOLL* | + | - | + | + |
| *iso; NimC1$^1$; Eater$^1$* | *Melcarne et al., 2019b* | *ΔPhag* | + | + | - | + |
| *iso; PPO1Δ, PPO2Δ; iso* | *Binggeli et al., 2014* | *ΔMel* | + | + | + | - |
| *+; +; Rel$^{E20}$, spz$^{rm7}$* | This paper | *ΔIMD, ΔTOLL* | - | - | + | + |
| *iso; PPO1Δ, PPO2Δ; NimC1$^1$,Eater$^1$* | This paper | *ΔPhag, ΔMel* | + | + | - | - |
| *iso; NimC1$^1$; Eater$^1$, Rel$^{E20}$* | This paper | *ΔIMD, ΔPhag* | - | + | - | + |
| *iso; PPO1Δ PPO2Δ; Rel$^{E20}$* | This paper | *ΔIMD, ΔMel* | - | + | + | - |
| *iso; PPO1Δ PPO2Δ; spz$^{rm7}$* | This paper | *ΔTOLL, ΔMel$^{PPS}$* | + | - | + | - |
| *iso Hay-psh$^{Def}$; iso; iso* | *Dudzic et al., 2019* | *ΔTOLL, ΔMel$^{HP}$* | + | - | + | - |
| *iso Hay-psh$^{Def}$; NimC1$^1$; Eater$^1$ Rel$^{E20}$* | This paper | *ΔITPM* | - | - | - | - |

six out of the seven combinations of flies simultaneously lacking two immune modules with the exception of *ΔTOLL, ΔPhag* as *NimC1[1]; eater[1], spz[rm7]* flies were not viable (see fly strains in *Table 1*). Of note, in our hands, recombinant *Rel[E20], spz[rm7]* flies were not viable in the DrosDel isogenic background, forcing us to use non-iso flies to assess the host defense of *ΔIMD, ΔTOLL* flies. Over the course of our study, isogenic *NimC1[1];eater[1]* flies also displayed viability issues, and specific experiments used non-isogenic flies to improve confidence in phenotypes; these experiments are noted in figure captions. We used two approaches to simultaneously delete the Toll and Melanization modules: (i) *PPO1[Δ], PPO2[Δ]; spz[rm7]* flies (referred to as *ΔTOLL, ΔMel[PPS]*) and (ii) flies carrying a small viable deletion *Hayan-psh[Def]* (referred to as *ΔTOLL, ΔMel[HP]*) that removes two clustered serine protease genes, *Hayan*, and *persephone*, which blocks the TOLL-PO SP cascade that regulates both humoral-Toll and melanization (*Dudzic et al., 2019*; *Westlake et al., 2024*). This allowed us to delete the Toll and melanization pathways using only one deletion event on the X chromosome. Excitingly, we succeeded in generating a fly line deficient for all four immune modules with the genotype: *Hayan-psh[Def]; NimC1[1]; eater[1], Rel[E20]* hereafter named *ΔITPM* (*ΔIMD, ΔTOLL, ΔPhag, ΔMel*). These flies were homozygous viable with very poor viability and homozygous fertile, despite major deficiencies in these four important systemic immune mechanisms. These flies allow us to monitor survival kinetics and pathogen growth in the near-total absence of an immune system. Finally, we added two isogenic fly lines in our analysis that delete key HDPs downstream of the Toll and Imd pathways: (i) *Bom[Δ55C]*, a short deletion removing ten Bomanins at position 55C, which causes a major susceptibility to infection by Gram-positive bacteria and fungi (*Clemmons et al., 2015*), and (ii) *ΔAMP14*, a compound fly line of eight mutations removing 14 AMP genes (4 Cecropins, 4 Attacins, 2 Diptericins, Drosocin, Defensin, Drosomycin, and Metchnikowin), which causes a major susceptibility to infection by Gram-negative bacteria (*Carboni et al., 2022*; *Hanson et al., 2019*). This set of immune-deficient fly lines in a defined genetic background provides a unique tool kit to address the function of immune modules individually or collectively.

## Imd, Toll, Phagocytosis, and Melanization modules function largely independently

The Imd, Toll, Phagocytosis, and Melanization modules are interconnected (*Westlake et al., 2024*). For instance, the Toll and Imd pathways both regulate a subset of genes involved in melanization (*De Gregorio et al., 2002*; *Ligoxygakis et al., 2002*). Moreover, one module can indirectly affect another. For instance, a lack of melanization could increase the bacterial load after infection, resulting in higher activation of the Toll and Imd pathways (*Binggeli et al., 2014*). As a first step, we analyzed the activation of each of the four modules in single- and multiple-module-deficient flies using classical readouts of these modules. This also allowed monitoring of the possible effects of deleting one module on the activity of another. We analyzed the immune response in single-module-deficient flies by monitoring (i) the expression of the antibacterial peptide gene *Diptericin* (*DptA* – a target of the Imd pathway), and (ii) the antifungal peptide gene *Drosomycin* (*Drs* – a target of the Toll pathway) upon systemic infection with a mixture of heat-killed *Escherichia coli* and *Micrococcus luteus*, (iii) ex vivo phagocytosis of *E. coli* and *Staphylococcus aureus* coated bio-particles incubated with larval hemocytes, and (iv) melanization at the injury site of adult flies. This analysis first confirmed that *ΔIMD, ΔTOLL, ΔPhag*, and *ΔMel* flies are deficient for Imd, Toll, phagocytosis, and melanization, respectively (*Figure 1A–D*). We note that the expression level of *DptA* was roughly similar to the wild-type in *ΔTOLL* and *ΔMel* flies; however, *DPhag* flies show reduced *DptA* induction at 6 hr compared to WT (~50%) (*Figure 1A*). This still represents an immense induction of *DptA* after infection but may suggest a lag or lesser overall Imd response in *DPhag* flies. *DptA* expression in *DPhag* was, however, comparable to wild-type at 12 and 24 hr. On the other hand, the *Drs* induction at 6 hr of *DMel* flies upon heat-killed *M. luteus* injection was significantly higher than WT (~1.6x), as found previously using live *M. luteus* infection (*Binggeli et al., 2014*; *Figure 1B*). The phagocytic assay confirmed that the *ΔPhag* mutant was significantly impaired in its ability to phagocytose bacterial bio-particles. Furthermore, *ΔIMD* and *ΔMel* mutant larvae had similar levels of phagocytic index to the wild-type (*Figure 1C*), while *ΔTOLL* had a higher phagocytic index, if anything. As expected, there was no apparent melanization at the site of the injury in *ΔMel* flies, while melanization blackening was readily observed in the wild-type and in other single-module mutants (*Figure 1D*). Collectively, this analysis indicates that each of the four immune modules can be selectively blocked in the isogenic DrosDel background without interfering with the competence of the others. We sometimes noticed a modest difference of module activity in

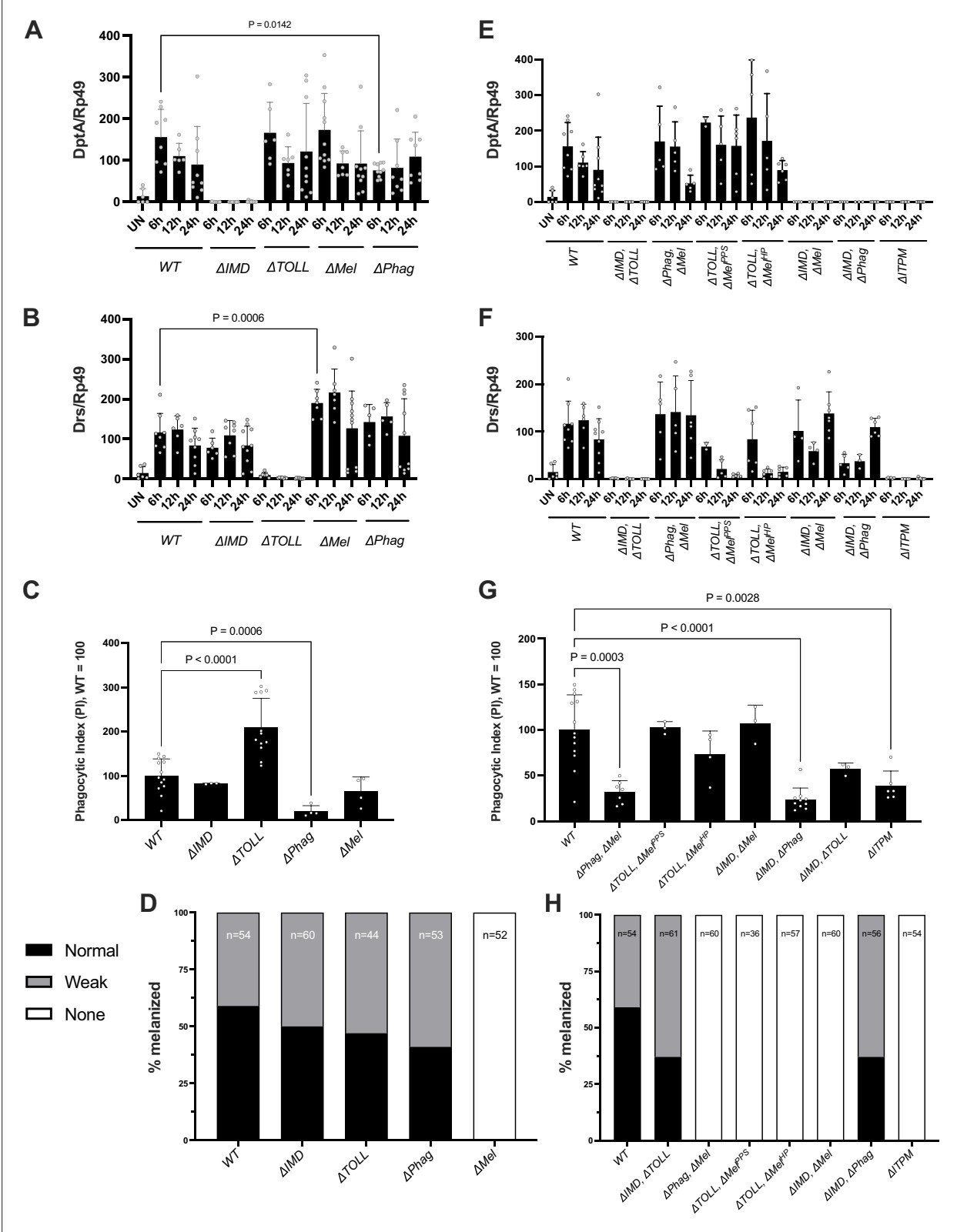

**Figure 1.** Each immune module functions largely independently from other immune modules. (**A**) Activation of the Imd pathway, as revealed by *DptA* expression following infection with a mixture of heat-killed *E. coli* and *M. luteus,* is broadly wild-type in single-module mutants other than Imd. In the case of *DPhag* flies, we observed a lesser induction at 6 hr compared to wild-type. (**B**) Activation of the Toll pathway, as revealed by the expression of *Drs*, is broadly wild-type in single-module mutants other than *DTOLL* flies. In the case of *DMel*, we observed a slightly greater Toll pathway activity. (**C**)

*Figure 1 continued on next page*

*Figure 1 continued*

The ability of plasmatocytes to phagocytose bacterial particles is not negatively affected in single-module mutants except *DPhag* (also see *Figure 1—figure supplement 1*). (**D**) Cuticle blackening after clean injury is not impaired except in *DMel* flies. (**E**) Activation of the Imd pathway remains strongly inducible in compound mutants except when deficient for the Imd pathway. (**F**) Activation of the Toll pathway remains strongly inducible in compound mutants except when flies were deficient of the Toll pathway. Note that *Drs* receives a minor input from Imd signaling (*Leulier et al., 2000*), explaining minor induction of *Drs* in *DToll* flies at early time points. (**G**) The ability of plasmatocytes to phagocytose bacterial particles is not significantly affected in compound mutants except those including *DPhag*. (**H**) Cuticle blackening after clean injury is not impaired in compound mutants except in *DMel* flies.

The online version of this article includes the following figure supplement(s) for figure 1:

**Figure supplement 1.** We used *DPhag* mutants in the genetic background of *Melcarne et al., 2019a* (*+; DPhag*) in some experiments, and confirm here that these mutants are equally deficient in phagocytic capacity to *DPhag* flies from the iso DrosDel genetic background (*DPhag*).

the absence of another (higher Toll activity in *ΔMel*, lower IMD activity in *ΔPhag*, and higher phagocytic activity in *ΔTOLL*), findings that require further exploration to be fully established.

We then repeated our analysis using our set of module compound mutants. Not surprisingly, all compound flies display the expected immune deficiencies validating the lines we constructed. This analysis confirms that *ΔITPM* flies, although viable, are indeed fully immune deficient for the four host defense modules. We did not detect any striking synergistic or antagonistic effects in which deleting two modules has a vastly different impact compared to the additive effect of deleting the two modules individually. Notably, a small level of *Drs* expression is retained in *ΔTOLL* mutant flies but was fully abolished in *ΔIMD, ΔTOLL* flies (*Figure 1A and E*), confirming the *Drs* gene receives a small input from the Imd pathway (*Leulier et al., 2000*). The *ΔTOLL, ΔMel^{PPS}* and *ΔTOLL, ΔMel^{HP}* flies display *Drs* induction at 6 hr, which is resolved to unchallenged WT levels by 12 hr, consistent with this induction coming from an early input from the IMD pathway in these backgrounds. Taken together, all the compound mutant flies behave largely as expected given the sum effects of their individual mutations, confirming that these modules largely function independently.

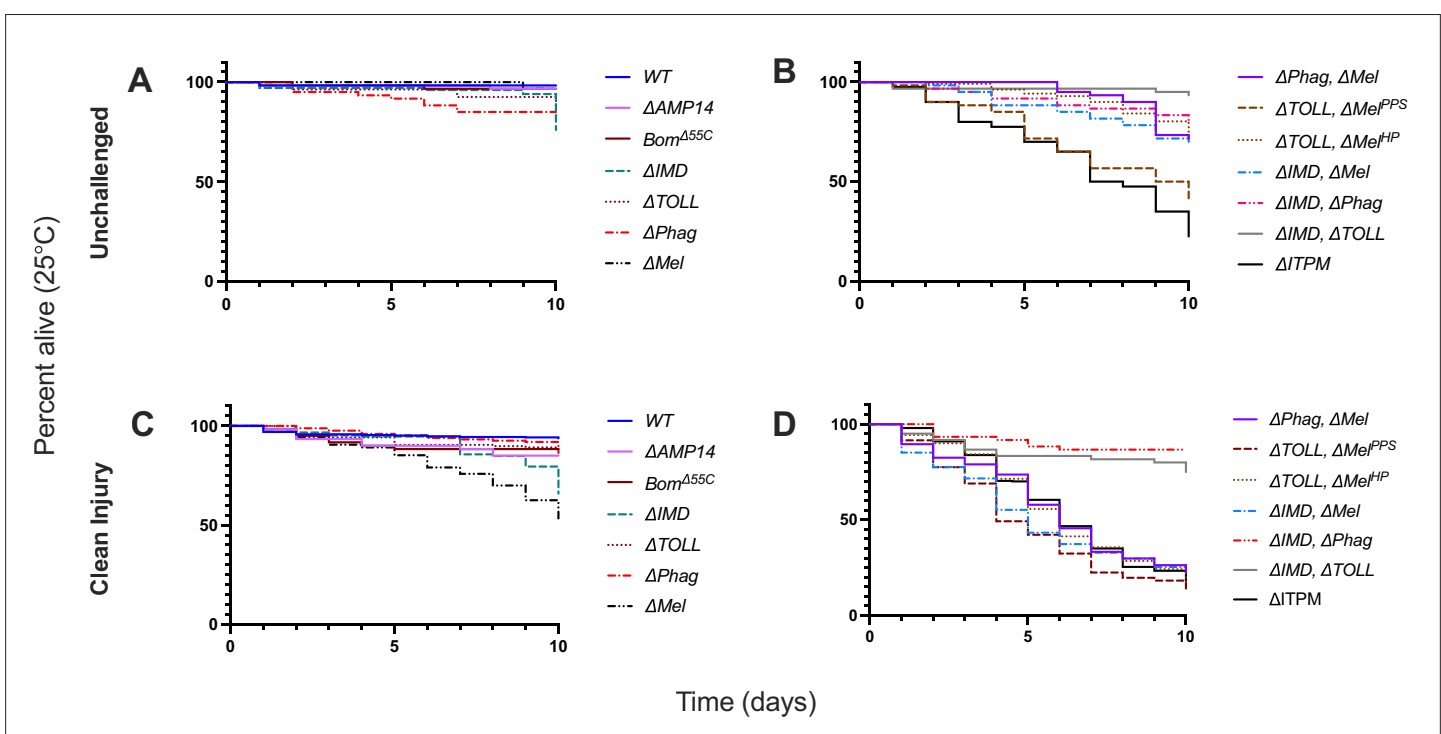

**Figure 2.** Lifespans of mutants used in this study in unchallenged flies (**A, B**) and upon clean injury (**C, D**) conditions at 25°C. See *Figure 2—figure supplement 1* for 29°C comparisons.

The online version of this article includes the following figure supplement(s) for figure 2:

**Figure supplement 1.** Lifespans of mutants used in this study in unchallenged flies(**A, B**) and upon clean injury (**C, D**) conditions at 29°C.

## Both the Melanization and TOLL modules contribute to wounding and lifespan

Before assessing the resistance to infection, we monitored fly survival at 25 and 29°C to determine if these mutations impact viability in the absence of challenge. At 25°C, no single mutants died significantly more than the wild-type in the absence of challenge within the 10-day window (*Figure 2A*). However, looking at compound mutants, even in the absence of challenge, each compound mutant other than ΔIMD, ΔTOLL suffered 25% cumulative mortality by 10 days post-eclosion, while ΔTOLL, ΔMel^PPS, and ΔITPM flies suffered even greater cumulative mortality. Similar trends were observed at 29°C with slightly elevated mortality (*Figure 2—figure supplement 1*). The observation that ΔTOLL, ΔMel^PPS (PPO1[1], PPO2[1], spz^rm7) flies have shorter lifespan compared to ΔTOLL, ΔMel^HP (Hayan-psh^Def) could be explained by residual Toll and/or PO activity in Hayan-psh^Def that preserves lifespan, or additional indirect effects in ΔTOLL, ΔMel^PPS flies deleterious to lifespan. As such, the combined effect of Toll and Melanization deficiencies in unchallenged and clean injury survival is best assessed by considering signals from both genotypes. ΔITPM flies (Hayan-psh^Def; NimC1[1]; Eater[1], Rel^E20) have the lowest unchallenged survival rate of genotypes included in this study. Taken together, the Imd, Toll, Melanization, and Phagocytosis modules contribute little to lifespan maintenance individually, but are crucial to maintain lifespan collectively (*Figure 2B*).

Systemic infections were performed by septic injury (pricked with a contaminated needle). We therefore monitored the survival of single and double mutant flies after a clean injury to disentangle the effects of injury from those of infection. We found a cumulative mortality of ~50% for ΔMel flies by day 10 (*Figure 2C*), while other single-module mutants displayed relatively little mortality over the 10-day window. DIMD had notable late-onset mortality in a minority of experimental replicates, which may be due to stochastic dysbiosis effects during aging (*Hanson and Lemaitre, 2023*; *Marra et al., 2021*). Double mutant ΔIMD, ΔTOLL and ΔIMD, ΔPhag flies retained a high survival rate. Consistent with an importance of melanization in the injury response (*Binggeli et al., 2014*), all the compound mutants affected in melanization were susceptible to clean injury, with similar mortality seen in ΔPhag, ΔMel, ΔIMD, ΔMel, ΔTOLL, ΔMel^PPS, ΔTOLL, ΔMel^HP, and ΔITPM flies (*Figure 2D*). Thus, while these genotypes differ in their base lifespan, their viability upon injury is remarkably similar. These results

| Type | Species | Temp | Dose | Days | N exp | WT | ΔAMP14 | ΔBom | ΔIMD | ΔTOLL | ΔPhag | ΔMel | ΔPhag, ΔMel | ΔTOLL, ΔMel^PPS | ΔTOLL, ΔMel^HP | ΔIMD, ΔMel | ΔIMD, ΔPhag | ΔIMD, ΔTOLL | ΔITPM |
|---|---|---|---|---|---|---|---|---|---|---|---|---|---|---|---|---|---|---|---|
| Controls | Unchallenged | 25°C | n/a | 7 | 13 | 6.9 177 | 6.9 60 | 6.9 60 | 6.8 60 | 6.6 60 | 6.6 60 | 7.0 116 | 7.0 60 | 6.0 60 | 6.8 239 | 6.6 60 | 6.6 60 | 6.8 60 | 5.8 80 |
| | Clean Injury | 25°C | n/a | 7 | 36 | 6.7 709 | 6.6 60 | 6.5 60 | 6.7 543 | 6.6 260 | 6.8 248 | 6.4 526 | 5.3 57 | 4.6 71 | 5.4 70 | 4.7 67 | 6.5 60 | 6.2 60 | 5.5 445 |
| | Clean Injury | 29°C | n/a | 7 | 11 | 6.6 211 | 6.5 60 | 5.9 60 | 6.3 100 | 6.1 75 | 6.6 60 | 6.4 119 | 4.9 96 | 3.0 67 | 4.8 56 | 3.7 57 | 6.0 64 | 5.6 60 | 3.9 94 |
| | Clean Injury | 25°C | n/a | 10 | 36 | 9.6 709 | 9.1 60 | 9.2 60 | 9.2 543 | 9.3 260 | 9.6 248 | 8.5 526 | 6.2 57 | 5.2 71 | 6.3 70 | 5.6 67 | 9.1 60 | 8.7 60 | 6.3 445 |
| | Unchallenged | 29°C | n/a | 10 | 11 | 9.9 139 | 9.3 60 | 9.5 60 | 8.9 60 | 7.1 34 | 9.2 60 | 8.7 60 | 5.9 64 | 9.3 40 | 8.5 60 | 9.1 60 | 9.4 49 | | 7.7 61 |
| Viruses | Drosophila C virus | 25°C | 2000 TCID50 | 10 | 5 | 8.3 100 | 6.7 89 | 7.7 85 | 6.8 73 | 7.8 83 | 8.1 78 | 6.3 86 | 6.3 88 | 5.5 56 | 5.0 92 | 5.1 60 | 6.7 64 | 3.8 53 | 4.1 78 |
| | Flock House virus | 25°C | 250000 TCID50 | 10 | 5 | 8.5 122 | 7.1 73 | 7.2 84 | 7.3 79 | 7.3 81 | 7.3 81 | 5.2 79 | 6.2 82 | 5.1 56 | 6.0 85 | 4.9 61 | 6.8 73 | 5.2 59 | 5.1 84 |
| | Drosophila X virus | 25°C | 431000 TCID50 | 10 | 2 | 9.0 40 | 6.9 41 | 8.2 20 | 7.1 45 | 7.9 34 | 7.8 41 | 5.4 38 | 7.4 46 | 4.9 37 | 5.1 38 | 6.0 15 | 6.2 40 | 5.6 19 | 4.6 43 |
| | Invertebrate iridescent-6 virus | 25°C | 31700 TCID50 | 10 | 3 | 9.4 59 | 7.5 50 | 9.4 64 | 8.1 58 | 8.4 54 | 8.8 60 | 5.6 58 | 6.0 61 | 5.8 59 | 5.8 58 | 6.1 48 | 8.3 54 | 5.9 47 | 4.4 69 |
| | Sindbis virus | 25°C | 50000 TCID50 | 10 | 3 | 9.4 58 | 8.4 61 | 8.6 61 | 8.9 60 | 8.8 60 | 8.3 60 | 5.4 45 | 5.9 66 | 4.9 51 | 4.8 59 | 4.5 60 | 7.0 50 | 4.5 45 | 4.3 61 |
| Fungi | Aspergillus fumigatus (natural) | 29°C | rolled on plate | 10 | 5 | 10.0 40 | 9.2 41 | 9.9 20 | 8.8 56 | 7.2 41 | 7.6 41 | 9.1 60 | 8.2 60 | 6.8 62 | 8.6 60 | 7.7 60 | 8.7 66 | 6.9 60 | 4.7 92 |
| | Beauvaria bassiana (natural) | 29°C | 30mg | 10 | 5 | 9.6 120 | 8.2 61 | 9.8 60 | 9.4 60 | 7.3 70 | 7.8 43 | 8.7 76 | 7.7 60 | 5.0 64 | 6.0 36 | 6.9 66 | 6.4 60 | 5.0 51 | 4.4 64 |
| | Beauvaria bassiana | 29°C | OD=5 | 7 | 6 | 6.4 41 | 6.0 74 | 3.6 59 | 5.3 58 | 3.6 66 | 5.2 39 | 5.3 62 | 4.4 57 | 2.6 69 | 3.4 63 | 4.4 41 | 4.7 51 | 3.1 56 | 3.3 95 |
| | Candida albicans | 29°C | OD=200 | 7 | 8 | 6.7 160 | 5.6 45 | 6.0 102 | 5.9 94 | 5.0 89 | 6.4 65 | 5.9 60 | 6.4 102 | 1.7 20 | 3.1 156 | 5.7 36 | 5.9 60 | 3.2 27 | 2.4 82 |
| Gram positive bacteria | Mycobacterium marinum | 25°C | OD=1 | 7 | 6 | 6.1 132 | 6.0 50 | 6.5 51 | 6.5 111 | 5.9 108 | 5.2 77 | 5.8 99 | 5.7 108 | 4.1 62 | 5.8 84 | 5.1 51 | 5.1 64 | 6.3 56 | 3.9 117 |
| | Corynebacterium diptheriae | 25°C | OD=100 | 7 | 3 | 6.0 73 | 6.2 84 | 6.5 97 | 5.6 79 | 5.1 60 | 6.6 87 | 4.6 82 | 5.8 69 | 3.2 47 | 3.3 103 | 4.9 72 | 5.9 88 | 4.5 61 | 3.0 75 |
| | Micrococcus luteus | 29°C | OD=200 | 7 | 8 | 6.7 121 | 5.7 60 | 6.8 60 | 6.2 120 | 6.4 148 | 6.5 71 | 6.5 87 | 5.6 81 | 4.6 66 | 6.5 80 | 4.8 77 | 6.3 84 | 2.1 102 | 2.1 91 |
| | Streptococcus pneumoniae | 25°C | OD=0.3 | 7 | 3 | 6.8 41 | 6.1 69 | 6.4 60 | 6.7 61 | 6.6 70 | 6.3 60 | 6.2 60 | 6.4 60 | 5.7 60 | 5.8 60 | 6.1 60 | 5.5 60 | | 4.5 50 |
| | Enterococcus faecalis | 25°C | OD=5 | 7 | 10 | 5.0 130 | 4.5 100 | 2.1 42 | 4.2 128 | 1.8 126 | 5.2 79 | 5.1 122 | 3.9 76 | 1.6 63 | 1.8 100 | 4.2 60 | 4.0 79 | 1.4 61 | 1.3 101 |
| | Staphylococcus aureus | 25°C | OD=0.5 | 7 | 8 | 5.1 119 | 4.8 58 | 5.7 21 | 4.2 118 | 3.4 73 | 3.6 79 | 4.2 60 | 1.9 78 | 2.0 80 | 2.0 112 | 2.1 78 | 4.1 59 | 3.4 40 | 1.9 87 |
| | Listeria monocytogenes | 25°C | OD=0.5 | 7 | 7 | 4.8 91 | 4.7 78 | 2.2 60 | 4.5 112 | 3.2 46 | 1.6 112 | 4.6 71 | 3.9 78 | 2.1 70 | 2.2 73 | 3.5 88 | 3.3 60 | 2.0 61 | 1.6 62 |
| | Bacillus subtilis | 25°C | OD=5 | 7 | 7 | 5.7 120 | 3.3 40 | 4.1 20 | 4.0 123 | 3.8 119 | 6.0 55 | 5.4 110 | 4.5 98 | 2.8 69 | 2.7 84 | 4.5 76 | 5.4 20 | 2.1 41 | 2.0 81 |
| Gram negative bacteria | Vibrio parahemolyticus | 25°C | OD=10 | 7 | 2 | 3.1 58 | 1.1 61 | 2.9 62 | 1.1 62 | 1.3 70 | 2.8 64 | 2.3 70 | 3.2 57 | 1.8 68 | 1.9 41 | 1.0 59 | 1.1 63 | 1.0 58 | 1.0 64 |
| | Providencia rettgeri | 25°C | OD=0.1 | 7 | 7 | 5.5 146 | 1.1 57 | 5.1 58 | 2.0 78 | 4.4 70 | 3.6 100 | 1.8 95 | 1.5 61 | 1.5 75 | 5.3 40 | 1.3 75 | 2.1 62 | 1.0 21 | 1.4 123 |
| | Providencia burhodogranaria | 25°C | OD=10 | 7 | 7 | 5.1 161 | 1.7 74 | 4.6 40 | 1.6 58 | 4.6 88 | 4.9 65 | 4.0 124 | 4.2 40 | 2.9 60 | 4.9 60 | 1.3 78 | 1.0 61 | 1.4 40 | 1.3 66 |
| | Pectobacterium carotovorum | 29°C | OD=10 | 7 | 8 | 6.4 151 | 1.5 57 | 5.8 40 | 1.8 111 | 5.8 60 | 4.4 120 | 5.8 100 | 5.2 102 | 4.7 60 | 5.2 60 | 1.8 72 | 1.8 61 | 1.4 62 | 1.4 100 |
| | Klebsiella pneumoniae | 25°C | OD=1 | 7 | 4 | 6.7 80 | 1.4 68 | 6.7 98 | 1.4 97 | 6.8 60 | 6.9 60 | 6.4 80 | 6.3 60 | 3.9 80 | 6.9 60 | 1.3 59 | 2.0 56 | 1.0 33 | 1.3 64 |
| | Enterobacter cloacae | 25°C | OD=10 | 7 | 8 | 6.6 149 | 1.3 80 | 6.7 60 | 1.0 100 | 6.6 90 | 6.7 60 | 6.4 60 | 6.1 80 | 4.9 61 | 5.8 60 | 1.3 73 | 1.0 37 | 1.6 61 | 1.2 83 |
| | Escherichia coli | 25°C | OD=200 | 7 | 8 | 6.6 141 | 5.0 79 | 6.9 40 | 3.8 80 | 6.2 70 | 5.4 20 | 5.5 43 | 5.5 85 | 3.9 63 | 5.3 63 | 3.8 62 | 4.6 60 | 1.7 70 | 1.6 60 |
| | Salmonella enterica ser. typhimurium | 25°C | OD=200 | 7 | 4 | 6.6 42 | 4.8 40 | 6.7 100 | 5.3 80 | 6.3 40 | 6.4 40 | 6.3 40 | 6.6 40 | 5.5 40 | 4.8 64 | 4.7 61 | 3.9 18 | 2.7 58 | 1.8 68 |

**Figure 3.** Heatmap of lifespans of immune module-deficient flies upon infection by various pathogens. Darker blue indicates lower survival, while white indicates maximum survival. Experiments used either 7 or 10 days as a maximum lifespan/time course, and the heatmap is adjusted accordingly per row to have white fill for the maximum possible lifespan of that row. Heatmap colors: blue (low survival), white (high survival). Survival curves are presented in *Supplementary file 2*, and a summary of susceptibilities is provided in *Figure 3—figure supplement 1*. Small numbers beside mean lifespans indicate total sample size per genotype per treatment.

The online version of this article includes the following source data and figure supplement(s) for figure 3:

**Source data 1.** Editable version of *Figure 3*.

**Figure supplement 1.** Summary table of susceptibilities per the three conditions outlined in section 'Systemic infections and survival'.

**Figure supplement 1—source data 1.** Editable version of *Figure 3—figure supplement 1*.

emphasize an interconnectedness of the melanization response with each other immune module for maintaining health after injury.

Similar results were obtained for both unchallenged and clean injury lifespans at 29°C. Of note, *DTOLL, ΔMel^{PPS}* suffered significantly greater mortality in unchallenged conditions compared to other genotypes, even including *DITPM* flies (*Figure 2—figure supplement 1*). Other compound mutant genotypes displayed a continuum of mortality in unchallenged conditions between *DITPM* (~50% cumulative mortality) and *ΔTOLL, ΔMel^{HP}* (~15% cumulative mortality). Upon clean injury, results were broadly consistent with 25°C trends, with a more uniform mortality across all genotypes.

Collectively, our study highlights that all these immune modules contribute somewhat to the survival of flies upon clean injury, with a major role of the melanization response in maintaining lifespan and survival to injury. In the next steps of our study, we compared survival upon infection to both unchallenged and clean injured flies to distinguish the impact of microbial infection from the injury itself.

## Multiple mechanisms contribute to resistance to microbial infection

To compare the contribution of each module to host defense, we performed infection experiments of wild-type and single- and multiple-module mutant flies following septic infection with five viruses, eight Gram-positive bacteria, eight Gram-negative bacteria, septic injury for two fungal species, and natural infection (i.e. spore deposition on the cuticle) for two fungal species. Temperature, dose, and days monitored are shown in *Figure 3*, and see *Supplementary file 1* for other key parameters. Survival curves are shown in *Supplementary file 2* for each pathogen and results of these survivals are summarized in *Figure 3* (and *Supplementary file 3*) as mean lifespans given 7 or 10 days as the maximum lifespan per treatment, with number of tested flies indicated on the right and darker blue intensity as indicative of lower lifespan.

We performed infection experiments with the following viruses and microbial species: DCV, FHV, DXV, IIV-6, SINV, *Aspergillus fumigatus, Beauveria bassiana, Candida albicans, Mycobacterium marinum, Corynebacterium diphtheriae, M. luteus, Streptococcus pneumoniae, Enterococcus faecalis, S. aureus, Listeria monocytogenes, Bacillus subtilis, Vibrio parahemolyticus, Providencia rettgeri, Providencia burhodogranariea, Pectobacterium carotavorum Ecc15, Klebsiella pneumoniae, Enterobacter cloacae, E. coli,* and *Salmonella enterica ser. typhimurium.* Collectively, this represents infections with 24 pathogens across 14 genotypes and two infection routes (336 genotype-by-pathogen interactions). When combined with comparisons to clean injury and unchallenged controls to validate if a lifespan difference is meaningful, and to wild-type or *ΔITPM* treatments, this inflates to 1000+ interactions. We therefore focused our survival analysis on major effects. We report the core summary statistics and overt trends and comment conservatively, only highlighting differences in mean lifespan between reference genotypes or control treatments with a minimum of 1 day.

Our study confirms that the Imd pathway plays a major role against all tested Gram-negative bacteria, as *ΔIMD* flies succumb more rapidly than wild-type flies to all of them. This pathway also contributes, albeit to a lesser extent, to survival against the Gram-positive bacteria *B. subtilis* and *L. monocytogenes,* which bear DAP-type peptidoglycan at their membrane that triggers Imd activation (*Kaneko et al., 2004*; *Leulier et al., 2003*). We were interested to see the impact of our *ΔIMD* flies in host defense to viruses as the specific mutation we used (*Rel^{E20}*) deletes a transcription factor common to both the canonical Imd pathway and the recently appreciated cGLR–Sting–Relish pathway (*Goto et al., 2018*). We found that *ΔIMD* mutant flies were somewhat susceptible to the four viruses DCV, FHV, DXV, and IIV6, but not SINV, agreeing with previous studies showing a reduction in lifespan of 1–2 days for *Rel* mutant flies after viral infection (*Costa et al., 2009*; *Goto et al., 2018*; *Sansone et al., 2015*).

This study also confirms a prominent role of the Toll pathway in survival upon infection by all virulent Gram-positive bacteria and fungi upon septic injury. Less virulent microbes and infectious routes (such as *A. fumigatus* by natural infection) did not result in increased mortality in individual *DTOLL* mutant flies. A contribution of the Toll pathway to survival was also observed for some Gram-negative bacteria, including a minor contribution to survival upon *Pr. rettgeri* infection, and a major contribution after *V. parahemolyticus* infection. In addition, we found a susceptibility of *ΔTOLL* flies to FHV, DXV, and IIV6, suggesting Toll-regulated effectors mediate defense against viral infection.

Use of *DPhag* mutants reveals a role of the cellular response in survival against the fungus *B. bassiana. DPhag* flies further show a minor susceptibility to *S. aureus* and a subset of Gram-negative

bacteria (*Pr. rettgeri*, *Ecc15*, and *E. coli*). Susceptibilities to Gram-negative bacteria could result from the observed lag in Imd pathway activation (*Figure 1A*); indeed, previous studies have shown a full *DptA* response is pivotal for defense against *Pr. rettgeri* (*Hanson et al., 2023*; *Hanson et al., 2019*; *Unckless et al., 2016*), and lifespans are greatly impacted by subtle differences when using pathogens with intermediate levels of virulence (*Duneau et al., 2017a*). A role for Imd in explaining *DPhag* susceptibilities does not detract from the importance of the cellular response, but instead offers a mechanistic route to explore for these phenotypes.

Surprisingly, our study reveals a relatively consistent role of melanization against all the tested viruses, notably so for DXV, IIV-6, and SINV. *ΔMel* flies also display a minor increased susceptibility to *B. bassiana* natural infection and septic injury. As previously described (*Dudzic et al., 2019*), *DMel* flies are highly susceptible to the Gram-positive bacterium *S. aureus*, rivaling the susceptibility of *DITPM* flies. *DMel* flies further show a strong susceptibility to *Pr. rettgeri* rivaling *DIMD* flies, and also a minor susceptibility to infection by other Gram-negative bacteria (*Pr. burhodogranariea*, *E. coli*).

Collectively, the use of single-module mutants confirms the major role of the Imd pathway against Gram-negative bacteria and Toll against virulent Gram-positive bacteria and fungi. Interestingly, melanization was consistently important to survive virus infection and plays a prominent and important role in defense against specific bacterial species. While it was almost never the most critical module, phagocytosis and the cellular response contribute to survival against various germs. Use of single-module-deficient flies alongside *DITPM* flies reveals that survival to some of the microbes tested can rely almost entirely on one pathway (Imd – *Ecc15*, *K. pneumoniae*, *E. cloacae*, *Pr. burhodogranariea*; Toll – *E. faecalis*), or can receive notable contributions from two pathways (Toll and IMD – *L. monocytogenes*, *B. subtilis*, *V. parahemolyticus*; Toll and Phagocytosis – *A. fumigatus* and *B. bassiana* natural infection; Imd and Melanization – DCV). Strikingly, many pathways contribute to host defense to *S. aureus* and *Pr. rettgeri*.

## Contribution of AMPs and Bomanins to Toll and Imd activities

Recent studies have described a prime importance of AMPs and HDPs in responding to infection (reviewed in *Hanson, 2024*). These peptides are the principal effectors of the Toll and Imd pathways, yet mutations for AMPs and HDPs typically have not been tested alongside mutations for both the Imd and the Toll pathways across pathogens, as studies have typically included just the one relevant pathway mutation as a positive control. Thus, the importance of AMPs and HDPs relative to the pathways that regulate them has been only partially investigated. We used *DAMP14* and *DBom* flies to assess the extent by which AMPs and Bomanins contribute to Toll and Imd activities (*Figure 3*). Here we found that AMPs are critical for survival to infection against all Gram-negative bacteria tested, often rivaling the susceptibility of *DIMD* flies. Notably, *DAMP14* flies were also somewhat susceptible to the yeast *C. albicans* and the Gram-positive bacteria *B. subtilis* and *M. luteus*, reinforcing minor susceptibilities seen previously (*Carboni et al., 2022*; *Hanson et al., 2019*). Interestingly, we found that *DAMP14* survivals to DCV, FHV, DXV, and IIV6 largely paralleled those of *DIMD* flies (*Figure 3*, *Supplementary file 1*), suggesting Imd-mediated survival phenotypes rely greatly on the presence of the seven classical AMP families.

On the other hand, *DBom* flies were susceptible to septic injury by *B. bassiana* and also somewhat to *C. albicans*, consistent with previous studies (*Supplementary file 2*), as well as all virulent Gram-positive bacteria, consistent with their important role downstream of the Toll pathway (*Clemmons et al., 2015*; *Lindsay et al., 2018*; *Xu et al., 2023*). In addition, *DBom* flies paralleled the *DTOLL* susceptibility to DCV, FHV, and DXV, but were not susceptible to IIV6 like *DTOLL*.

Collectively, our study confirms the key role of AMPs downstream of the Imd pathway in the defense against Gram-negative bacteria, and a primary role of Bomanins downstream of Toll in defense against fungi and Gram-positive bacteria, with little input to survival against Gram-negative bacteria. We additionally find these secreted peptides can contribute to survival after certain viral infections.

## Immune modules mostly contribute additively to host defense

The results described above revealed the key contribution of single modules and immune effectors to host defense. However, they did not assess the possible additive, synergistic, or antagonistic contribution of these four modules. Moreover, they did not monitor to which extent these modules were

important in the absence of other modules. To address this, we compared the survival of single- to double-module-deficient flies lacking two modules and *DITPM* flies lacking all four modules (*Figure 3*).

We first observed that *ΔITPM* flies, which lack the four modules, were always as or more susceptible than single- and double-module-deficient flies. In multiple cases, single or double mutants already rivaled the susceptibility of *ΔITPM* flies in microbe-specific ways. In particular, the susceptibility of *ΔIMD* flies to most Gram-negative bacteria rivaled *ΔITPM*, and for some germs, *DTOLL* (*L. monocytogenes*, *E. faecalis*, *B. bassiana* injury) or *DMel* (*S. aureus*) alone was comparable to *DITPM*. In other cases, however, combined loss of two pathways was necessary to rival *DITPM* susceptibilities. For instance, *ΔIMD, ΔTOLL* deletion causes complete *DITPM*-like susceptibility to *B. subtilis*, *M. luteus*, and *E. coli*, and near-complete susceptibility to *Sa. typhimurium*. *ΔIMD, ΔTOLL* further causes increased susceptibility to most viruses compared to their individual mutant strains. In these cases, there is clear synergy or additivity of the contributions of the two pathways, as either pathway alone results in only a minor or no susceptibility. We also observed intriguing differences between our two versions of *ΔTOLL, ΔMel* flies, as *ΔTOLL, ΔMel^{PPS}* tended to succumb to infection to a greater extent than *ΔTOLL, ΔMel^{HP}*, consistent with the differences observed upon clean injury (*Figure 2*). *DTOLL, DMel^{PPS}* displayed a minor increase in susceptibility compared to *DTOLL, DMel^{HP}* to *A. fumigatus* natural infection, and more prominent susceptibilities to the yeast *C. albicans*, the Gram-positive bacteria *M. marinum*, *M. luteus*, and also to Gram-negative bacteria *K. pneumoniae*, *E. coli*, *Pr. rettgeri*, and *Pr. burhodograneriea*.

Collectively, use of double-module-deficient flies uncovers the role of immune modules to host defense that were masked by other modules. While *DIMD* and *ΔTOLL* deficiencies sometimes displayed a synergistic effect (i.e. little mortality of single mutants, but *DITPM*-like mortality of double mutants), for combinations of other pathways, the results were more often minor with a cumulative effect on susceptibility, suggesting most modules contribute to defense independently. The use of two variations of *DTOLL, ΔMel*, with *ΔTOLL, ΔMel^{PPS}* having a stronger impact than *ΔTOLL, ΔMel^{HP}*, also highlights how the level of pathway disruption can affect infection outcomes, particularly for certain fungal or bacterial pathogens like *Providencia* species that may cleave host proteases (suggested by *Duneau et al., 2017b*; *Issa et al., 2018*).

## Timing of Toll, Imd, Phagocytosis, and Melanization contribution to host defense

Host defense programs can rely on resistance mechanisms that directly target or limit growth of pathogens (*Howick and Lazzaro, 2017*; *Duneau et al., 2025*; *Medzhitov et al., 2012*). When effectors reach a critical concentration threshold that promotes resistance, pathogen growth is inhibited; for instance, the time to pathogen control of *Pr. rettgeri* is ~7 hr, relying heavily on the expression and production of DptA (*Duneau et al., 2017a*; *Hanson et al., 2023*). Phagocytosis and the melanization response are often described as providing near-immediate immune protection (*Haine et al., 2008*), with reactions ex vivo progressing within minutes. Meanwhile, activation of the Imd and Toll pathways takes hours to reach peak concentrations of HDPs (*Uttenweiler-Joseph et al., 1998*). Thus, we were curious if different mutations would display differences in their time of action to control microbial growth. We therefore measured the microbial growth rate for *Pr. rettgeri*, *S. aureus*, and *C. albicans* at different time points post-infection in wild-type and single-module mutant flies as these pathogens were combatted by several immune modules in our survival data.

Agreeing with contributions to survival, flies singly deficient for IMD, Phagocytosis, or Melanization modules display a higher bacterial load than wild-type when infected with *Pr. rettgeri* (*Figure 4A*), which was apparent after 12 hr post-infection. Our observation that *DPhag* showed impaired IMD signaling at 6 hr (*Figure 1A*) might explain the contribution of phagocytosis to *Pr. rettgeri* microbial growth control with the same timing as *DIMD* flies. We observed that *DIMD* flies, *DMel* flies, and *DITPM* flies that each succumb completely to *Pr. rettgeri* infection also show high and consistent microbe loads at 24 hr indicative of all individuals progressing toward sepsis-induced death. On the other hand, wild-type, *DTOLL*, and *DPhag* flies show a greater stochasticity in bacterial load at 24 hr, consistent with a fraction of individuals surviving the infection by controlling the pathogen. This indicates deletion of the Toll pathway, and phagocytosis does not fully ablate essential resistance mechanisms for combatting *Pr. rettgeri*.

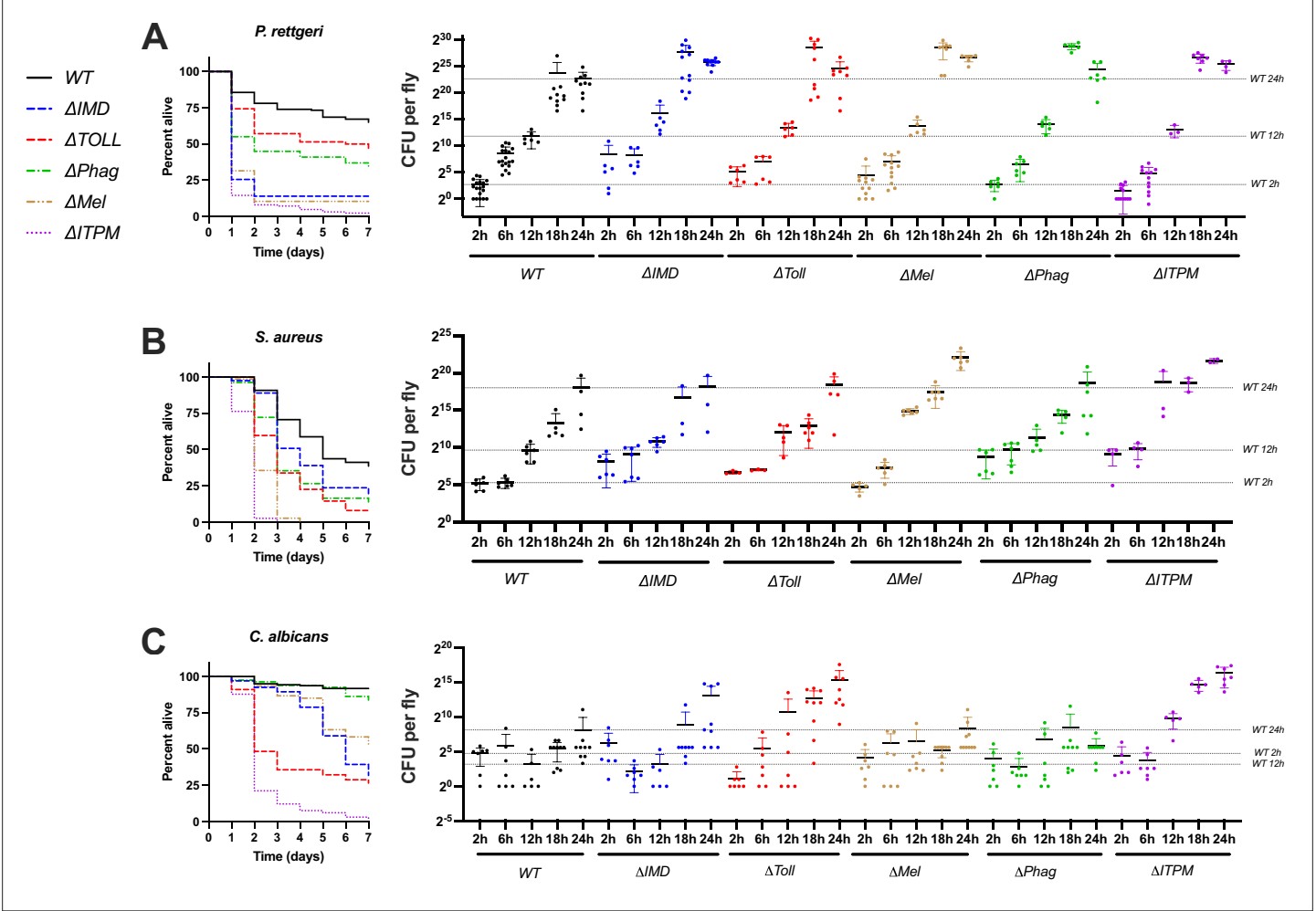

**Figure 4.** Growth kinetics of *Pr. rettgeri* (**A**), *S. aureus* (**B**), and *C. albicans* (**C**) in wild-type, single-module mutant and *DITPM* flies. Survival curves underlying data in *Figure 3* are shown for comparison. Each data point reflects a pooled sample of five flies. Error bars reflect 1 standard deviation from the mean.

The online version of this article includes the following figure supplement(s) for figure 4:

**Figure supplement 1.** Survival to infection of females against *P. rettgeri*, *S. aureus*, and *C. albicans*.

In the case of *S. aureus* (*Figure 4B*), we observed some individuals with higher bacterial loads in *ΔPhag* and *ΔIMD* flies at 2 and 6 hr, which was not seen in wild-type, *ΔTOLL*, or *ΔMel* flies. This suggests the Imd pathway can suppress early *S. aureus* growth, and possibly also phagocytosis, although this *ΔPhag* could ultimately stem from lesser Imd induction. Interestingly, flies lacking melanization displayed the highest susceptibility but did not depart from wild-type or other modules in pathogen load until 12 hr post-infection. While the melanization reaction is a rapid response, these results suggest that the killing activity of the melanization response acts with slower kinetics. In *DITPM* flies, we further observed earlier *S. aureus* growth at 2 and 6 hr like *DIMD* and *DPhag* flies. However, *S. aureus* loads at 12–24 hr were more comparable in *DITPM* flies to *DMel* flies. These trends agree both with survival data kinetics and an independent action of each of these pathways in their resistance effects against *S. aureus*. This suggests that *DITPM* flies succumb even more quickly than individual modules due to loss of multiple resistance mechanisms with different kinetics of activation that independently contribute to defense.

For *C. albicans* (*Figure 4C*), mortality begins 2–3 days post-infection for *ΔTOLL* mutants, but takes place at later time points for *ΔIMD*, *ΔPhag*, and *DMel* flies. In microbial growth kinetics, *ΔTOLL* already shows high *C. albicans* loads at 24 hr consistent with a previous study (*Hanson et al., 2019*) and observations done with *Candida glabrata* (*Quintin et al., 2013*). Interestingly, a

few *ΔIMD* flies also showed elevated growth of *C. albicans* at early time points. This suggests Imd-regulated effectors could help to suppress initial *C. albicans* growth, while Toll-regulated genes (e.g. Bomanins, Drs) contribute to this pathogen suppression to a greater extent at later time points. On the other hand, *ΔPhag* and *ΔMel* flies did not show any increased *C. albicans* loads within the 24 hr time window despite onset of mortality at later time points. Finally, *ΔITPM* flies displayed more consistent and rapid growth of *C. albicans* than *DTOLL* or *DIMD* alone, particularly visible at 12–24 hr post-infection, consistent with independent contributions of both Toll and Imd in resistance to this yeast.

All survival experiments to this point were done with males. We therefore assessed key survival trends for these infections in females to learn whether the dynamics we observed were consistent across sexes (*Figure 4—figure supplement 1*). For all three pathogens (*Pr. rettgeri*, *S. aureus*, and *C. albicans*), the rank order of susceptibility was broadly similar between males and females, with higher rates of mortality in females overall. Thus, we found no marked sex-by-genotype interaction. Interestingly, the greater susceptibility of females in our hands is true even for *ΔITPM* flies against *C. albicans*, although there were only a few surviving flies on which we can base these conclusions. However, these data may suggest the sexual dimorphism in defense against infection that we see against these pathogens is due to factors independent of the immune modules we disrupted.

Collectively, we observed that microbial growth parallels the susceptibility of modules indicating that they work in resistance. Surprisingly, we did not observe an early role in pathogen killing for melanization despite melanization regulating a rapid blackening response in ex vivo assays (*Nakhleh et al., 2017*). We further recover stochasticity in microbe loads across genotypes as expected when only a subset of individuals suppresses the pathogen and survives the infection (*Duneau et al., 2017a*). The more rapid and more consistent microbial growth kinetics of *DITPM* flies, particularly visible by 18 hr, further demonstrate that these pathways contribute to resistance collectively.

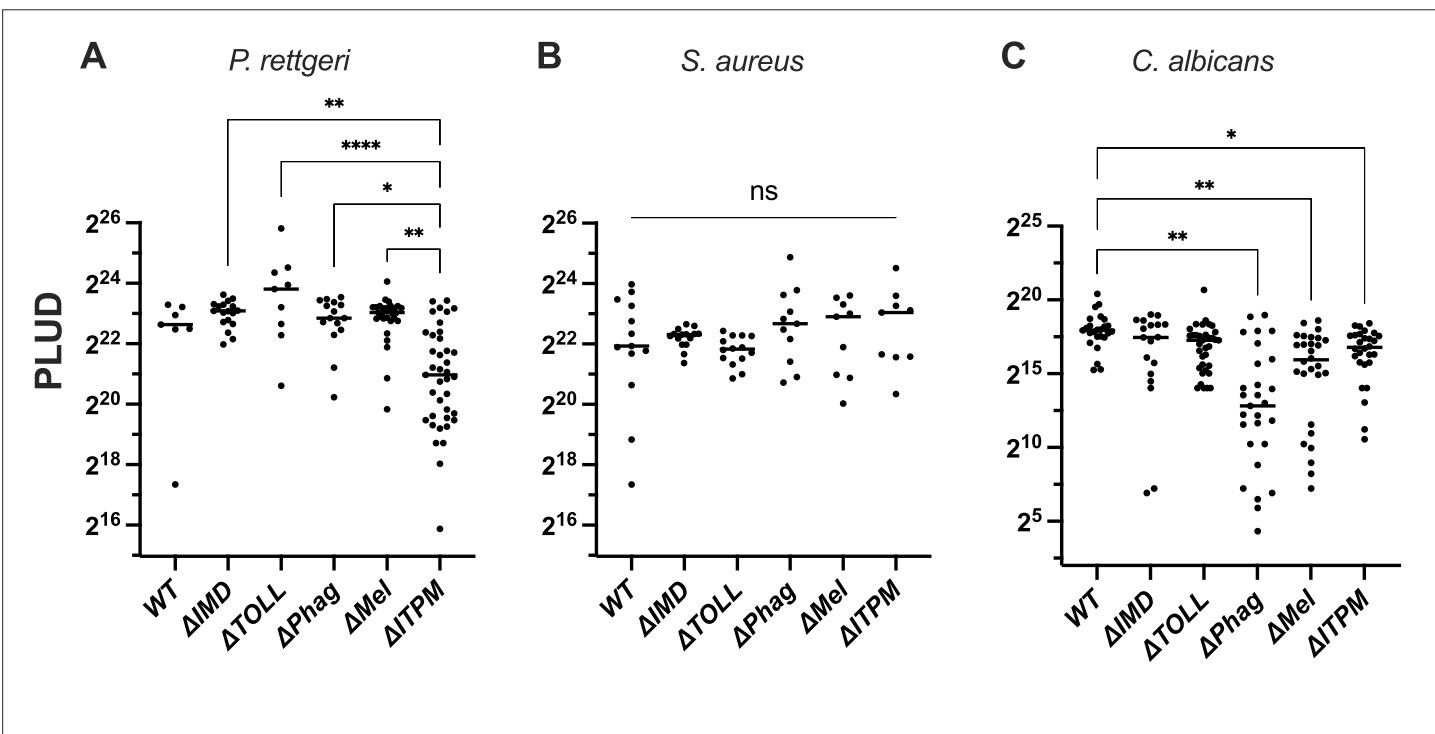

**Figure 5.** Measurement of Pathogen Load Upon Death (PLUD) upon infection with *Pr. rettgeri*, *S. aureus*, and *C. albicans*. (**A**) The PLUD of *Pr. rettgeri* in individual module mutant flies is not significantly different from wild-type. The PLUD of *DITPM* flies was not significantly different, although censoring of a single low-PLUD outlier in the wild-type would result in a significant difference between wild-type and *DITPM* flies (p < 0.01). (**B**) The PLUD of *S. aureus*-infected flies is not different across any genotype. (**C**) The PLUD of *C. albicans*-infected flies was significantly lower in *DPhag* flies with a notably lower mean PLUD. This difference was robust to use of different *DPhag* genetic backgrounds (merged data shown here). *DMel* flies and *DITPM* flies also had significantly lower PLUD. Error bars reflect 1 standard deviation from the mean. * = P < .05, ** = P < .01, *** = P < .001.

## Contributions of individual modules to disease tolerance

An increasingly used metric to delineate roles of resistance and tolerance is the Pathogen Load Upon Death (PLUD) (*Duneau et al., 2017a*; *Duneau et al., 2025*). The PLUD is determined by the virulence of the pathogen to set an upper limit, but also the tolerance of the host in surviving to a given pathogen burden before succumbing to infection. We monitored PLUDs for the same pathogens: *Pr. rettgeri, S. aureus*, and *C. albicans*. The PLUD of both *Pr. rettgeri* and *S. aureus* upon death was overall similar in wild-type and single-module mutant flies (*Figure 5A, B*). Notably, there was a greater stochasticity in *Pr. rettgeri* infections trending toward lower PLUD in *ΔITPM* flies, suggesting a stochastic reduction in disease tolerance of *DITPM* flies to this bacterium, although this was not significant compared to wild-type ($p > 0.05$). We saw no statistical differences in PLUD across genotypes for *S. aureus*.

For *C. albicans*, the distribution of PLUD values for *DPhag* flies was markedly different from wild-type, also seen to some extent for *DMel*. Interestingly, flies with these genotypes suppress *C. albicans* growth well (*Figure 4C*). Yet here we recovered many individuals from both of these module-deficient lines that died with a far lower PLUD (*Figure 5C*). Taken together, this suggests that the humoral immune pathways, particularly Toll, contribute to resistance against *C. albicans*. However, the cellular response and melanization reaction instead regulate tolerance to *C. albicans* infection. The use of *ΔITPM* flies further emphasizes that when both resistance (Toll, Imd) and tolerance (Phagocytosis, Melanization) mechanisms are deficient, the loss of resistance plays primary importance and can mask the loss of tolerance from being observed.

Collectively, monitoring of PLUD suggests that some modules, specifically phagocytosis and melanization, contribute to tolerance of certain infections. Importantly, this tolerance effect was only seen in genotypes that can resist infection. These findings demonstrate the collective contribution to resistance and/or tolerance of the four immune modules studied here.

## Discussion

In this article, we have generated a set of single and compound immune module-deficient fly lines in a controlled genetic background. Using various assays, we confirmed the validity of our lines, revealing that each module can be activated independently. We did, however, observe a higher Toll activation in *ΔMel* flies upon systemic infection with dead bacteria. Future studies may reveal if this higher Toll activity in *ΔMel* flies involves a regulatory pathway, or perhaps reflects higher persistence of microbial elicitors (e.g. peptidoglycan) that activate the Toll pathway. We also observed a lesser Imd activation in *ΔPhag* flies at 6 hr post-infection, which could reflect a role for hemocytes to stimulate the Imd systemic response. In this study, we used *NimC1[1]; eater[1]* double mutants to assess the cellular response. These flies have defects in phagocytosis, but also adhesion and sessility (*Melcarne et al., 2019b*). *NimC1[1]; eater[1]* larvae also have increased hemocyte number at the larval stage, but adults rapidly lose Hml-positive cells (*Melcarne, 2020*). Despite these limitations, we believe that the mutations we have used here are among the best available to assess the role of these four modules to host defense. Surprisingly, we could produce a fly line lacking the four main defense mechanisms of the systemic immune response, indicating that none of these modules is essential for survival. Theoretically, *ΔITPM* flies are almost completely immune deficient, but they can still clot wounds, activate the JNK and JAK–STAT pathways that mediate the wound healing response, and retain constitutive immune defense molecules that could also provide a certain degree of protection. Consistent with several studies (*Capilla et al., 2017*; *Carvalho et al., 2014*; *Rämet et al., 2002*), we show that Toll and melanization synergistically contribute to wound healing in adults. However, the observation that *ΔPhag, ΔMel* or *ΔIMD, ΔMel* double mutant flies are also more susceptible to clean injury than *ΔMel* flies indicates that the Imd pathway and phagocytosis also contribute to wound healing. Thus, although immune-deficient fly lines for the four modules are viable, the use of compound mutation and *ΔITPM* flies reveals a clear role of the immune system in response to wounding and lifespan maintenance.

Use of single- and double-module-deficient flies provides key insights on the mechanisms used by *Drosophila* to combat infection. We could confirm previous studies revealing the role of Imd against Gram-negative bacteria, Toll against Gram-positive bacteria and virulent fungi, and an importance of melanization and phagocytosis against specific pathogens (*Charroux and Royet, 2009*; *Defaye et al., 2009*; *Garg and Wu, 2014*; *Nehme et al., 2011*). Surprisingly, melanization was a consistently

important module in survival to virus infection. It seems unlikely that PPO activity taking place in the hemolymph could combat viral agents that are intracellular. We instead speculate this contribution of melanization to surviving virus infection could be due to its role in wound healing, a role in infected cell clearance, or perhaps autotoxic contributions of melanization reaction intermediates that fail to convert in phenoloxidase-deficient flies. We additionally recovered a role of Imd signaling (i.e. *Relish*) in antiviral defense, which was expected given recent characterizations of cGLR–Sting–Relish antiviral immunity (*Ai et al., 2024*; *Cai et al., 2022*; *Goto et al., 2018*). However, susceptibilities of *Rel^{E20}* flies were paralleled by *ΔAMP14* in all cases, suggesting AMPs contribute to this susceptibility. While Sting regulates a number of genes likely important for antiviral defense (*Goto et al., 2018*), the susceptibility we observe here could be a direct action of AMPs on enveloped viruses as described in some studies (*Feng et al., 2020*; *Huang et al., 2013*; *Yasin et al., 2004*), or an indirect effect, such as a failure to regulate gut microbes after the damage induced by viral replication (*Marra et al., 2021*). We additionally used dual modes of infection for the fungus *B. bassiana*, finding an importance of melanization and Toll signaling in both infection modes. However, our study reveals that Bomanin effectors explain most of the Toll contribution upon septic injury but not natural infection, for *B. bassiana*. This observation is in line with different importance of effectors or modules according to the route of infection (*Martins et al., 2013*).

Our double-module mutant analysis reveals that most modules contribute additively to host defense. This is consistent with these modules functioning independently. However, we noticed instances of synergy between two pathways, notably including Toll and Imd. Synergy between Toll and Imd can be explained by the fact that many immune-inducible regulated genes receive input from both the Imd and Toll pathways, including genes like *Metchnikowin*, *Drosomycin*, and *Transferrin1* (*De Gregorio et al., 2002*). Finally, we also observed rarer cases of synergistic susceptibility in flies deficient for Toll and Melanization or Imd and Melanization.

Our study confirms that the Toll and Imd humoral modules provide a broad role against certain classes of pathogens, Imd for Gram-negative bacteria and DAP-type containing Gram-positive bacteria, and Toll for Fungi and Gram-positive bacteria. Use of AMP and Bomanin mutants revealed that this can be largely explained by the effectors they control. In contrast, the contribution of phagocytosis and melanization appears to be critical to a more specific and diverse set of pathogens. We speculate that phagocytes or melanization are critical to handle bacteria that resist HDPs (*Hanson et al., 2019*), or can hide from them (*Touré et al., 2023*). The melanization reaction is a source of ROS that is potent against pathogens such as *S. aureus* that have been shown to be sensitive to ROS and resistant to Toll and Imd defenses (*Dudzic et al., 2019*; *Ford and King, 2021*; *Needham et al., 2004*; *Ramond et al., 2021*). These two modules play important roles in immune-related processes such as encapsulation (melanization), the uptake of bacteria escaping from the gut, or tissue homeostasis (phagocytosis) (*Braun et al., 1998*; *Melcarne et al., 2019b*; *Nehme et al., 2007*), which were not assessed here. Collectively, our study validates, with minor discrepancies, many studies that have assessed the contribution of these modules individually (e.g. *Apidianakis et al., 2005*; *Binggeli et al., 2014*; *Lamiable et al., 2016*; *Lemaitre et al., 1996*; *Lemaitre et al., 1995*; *Nehme et al., 2011*). Our mutant lines can now be used to analyze the contribution of these immune modules in resistance to other pathogens, notably wasps, nematodes, microsporidia, and protozoans, or in other contexts such as mating and local infection.

While some immune modules play a predominant role against some pathogens, other pathogens are handled by multiple modules. We hypothesize that pathogens that have intermediate levels of virulence, killing only a fraction of wild-type flies, may better reveal the role of multiple modules. Indeed, the stochasticity in survival analysis partly stems from the arms race occurring between the pathogen and host immunity, as shown for *Pr. rettgeri* (*Duneau et al., 2017a*). In this condition, any small change in the immune system may tip the outcome of the arm race toward lethality or survival. In this line of thinking, it is notable that multiple modules were important to survive *Pr. rettgeri* infection. Previous studies have revealed a major role of the Imd pathway-regulated AMP Diptericin A against this bacterium (*Hanson et al., 2023*; *Hanson et al., 2019*; *Unckless et al., 2016*). However, Duneau et al. showed that survival patterns to *Pr. rettgeri* bifurcate into two outcomes based on time taken to fully activate systemic defenses (*Duneau et al., 2017a*), and showed a role for a Toll-PO SP cascade regulating serine protease in defense against this microbe (*Duneau et al., 2017b*). We may speculate that melanization, although not as uniquely critical as Diptericin, might tip the balance of this arms

race toward host lethality, resulting in comparable survival phenotypes. The observation that *DPhag* flies expressed less *DptA* (*Figure 1*) can also explain their susceptibility. Future studies will clarify how these multiple modules can contribute to host survival according to additive or Achilles dynamics – the concept that microbes have generic or specific weaknesses that host effectors can target (*Hanson, 2024*). It will also be important to consider the distinct roles of resistance, tolerance, and resilience in host defense (*Howick and Lazzaro, 2017*; *Wukitch et al., 2023*; *Duneau et al., 2025*).

We observed a good correlation between survival analysis and pathogen growth in single-module flies for *Pr. rettgeri*, *S. aureus*, and *C. albicans*. This indicates that these immune modules mostly contribute to resistance mechanisms that target pathogen growth. Our study did not reveal key early contributions of phagocytosis or melanization to control pathogen growth, despite their quasi-immediate activation; higher *S. aureus* growth at 2 hr in *DPhag* flies was paralleled by *DImd* flies, and we observed lesser Imd activation in *DPhag* flies. Melanization, while being critical to survive *S. aureus* infection, impacts bacterial growth beginning only after the 6-hr time point. This indicates that the melanization microbicidal activity in vivo takes place slower than the blackening reaction seen from bled hemolymph. Interestingly, *ΔPhag* and also *ΔMel* flies could suppress *C. albicans* yeast growth, but ultimately some individuals succumb to infection with lower PLUD levels. We confirmed *ΔPhag* PLUD results using both an isogenic and a second wild-type genetic background, suggesting this lower PLUD is genuine. Susceptibility to fungal infection independent of fungal proliferation has also been reported using an *A. fumigatus* septic infection model and relies on the protection offered by Bomanins from pathogen-secreted toxins (*Xu et al., 2023*). It is tempting to speculate, based on the modules involved, that the loss of tolerance to *C. albicans* we observe has to do with wound repair or accumulating damage, such as what has been reported in beetles that suffer renal failure (*Khan et al., 2017*; *Li et al., 2020*), or flies with autotoxic trachea degradation upon stress pathway disruption (*Rommelaere et al., 2024*). Thus, this tolerance effect could rely on pathogen-mediated or autotoxic damage, which may be elucidated in a future study.

Here, we have provided a single and double mutant analysis of *Drosophila* immune module functions. Our study provides several insights on what modules are most important to survive infection by defined pathogens. However, we also highlight the collective contribution of modules to defense even when one module is of an outsized importance. We extend our comprehension of innate immune responses by revealing higher complexity, implicating multiple host defense modules in survival to various germs, including some with more cryptic contributions. As illustrated by our previous characterizations of AMP function (*Carboni et al., 2022*; *Hanson et al., 2019*), the melanization response (*Dudzic et al., 2015*), and stress-induced Turandot proteins (*Rommelaere et al., 2024*), a combinatorial mutation approach to deciphering immune functions can be extended even to the broad level of immune modules. Of note, we were unable to systematically sample all genotype-by-pathogen interactions equally. We have therefore been highly conservative in our reporting of major effects. There are likely many important interactions not discussed in our study. Future investigations may highlight important biology that is apparent in our data, but which we may not have mentioned here. To this end, we have deposited our isogenic immunity fly stocks in the Vienna *Drosophila* Resource Centre to facilitate their use. Beyond immunity, our tools can also be of use to study various questions at the cutting edge of aging, memory, neurodegeneration, cancer, and more, where immune genes have been implicated repeatedly. We hope that this set of lines will be useful to the community to better characterize the *Drosophila* host defense.

## Materials and methods

### Insect stocks

Precise details of *Drosophila* stocks used in this study are provided in *Table 1*. To minimize the influence of genetic background for the mutations used in this study, mutations were partially isogenized into the *DrosDel iso w[1118]* genetic background (BDSC #5905) as described by *Ferreira et al., 2014*; *Ryder et al., 2004*. Isogenized lines were generated and used for every mutant except the double mutant Toll and Imd (*Rel[E20], spz[rm7]*), for which homozygous isogenized flies were not viable in our hands. Double mutants for *NimC1[1]; Eater[1]* were poorly viable when isogenized in the *DrosDel iso w[1118]* background, and so we used both isogenic and non-isogenic lines over the course of our study to facilitate investigation. Furthermore, the *NimC1[1]; spz[rm7], Eater[1]* triple mutants were homozygous

lethal isogenized or not, and so could not be included. In addition, the line *ΔAMP14* bears a knockout of 14 different AMP genes, which includes the AMP families Drosomycin, Metchnikowin, Cecropin, Defensin, Drosocin, Attacin, and Diptericin, described previously (*Carboni et al., 2022*; *Hanson et al., 2019*). The *Bom^D55C* mutation was characterized in *Clemmons et al., 2015* and isogenic flies generated in *Hanson et al., 2019*. For larval experiments, the use of GFP-labeled CyO or TM6,Tb balancers was used to enable picking of homozygous mutant larvae.

Of note, the strains used in this study differ in their presence/absence of the *white^+* gene, present in the *PPO1^Δ*, *NimC1^1*, and *eater^1* mutations. In addition to its well-established function in eye pigmentation, the *white* gene can also impact host neurology and intestinal stem cell proliferation (*Ferreiro et al., 2017*; *Sasaki et al., 2021*). We did not observe any obvious correlations between *white^+* gene status and susceptibilities in this study. Moreover, in a previous study looking at the cumulative effects of AMP mutations on lifespan, *white* gene status and fluorescent markers did not readily explain differences in longevity (*Hanson and Lemaitre, 2023*). We therefore believe that the extreme immune susceptibility we have created through deficiencies for pathways regulating hundreds of genes, or major immune modules, overwhelms the potential effects of *white^+* and other transgenic markers. For additional information on which stocks bear which markers, see discussion in *Supplementary file 4*.

## Microorganisms culture

Microbe strain information, microorganism classifications, and culture conditions are listed in *Supplementary file 1*. Bacterial cultures and *C. albicans* were grown overnight shaking at 200 rpm. *A. fumigatus* fungus was grown on Malt Agar at room temperature until sporulation. The entomopathogenic fungus *B. bassiana* strain R444 was provided by Andermatt AG as spore preparations (BB-PROTEC) which were directly used in natural infections or dissolved in PBS for septic injuries. Viruses DCV and FHV were kindly provided by Carla Saleh (Pasteur Institute, Paris), produced in S2 cells. The supernatant of S2 cells was titrated before being used for fly infections. Viruses SINV, DXV, and IIV-6 were kindly provided by Ronald van Rij (Radboud University Medical Center, Nijmegen).

Heat-killed microbes were prepared by two repeats of boiling at 95°C for 30 min then freezing at –20°C for 30 min, before storage long term at –20°C. Microbe preparations were streaked onto agar plates to check for full efficiency of heat-killing before use in experiments.

## Gene expression

Flies were inoculated by pricking in the junction of thoracic pleura with a needle dipped in a mixed pellet containing a 1:1 mixture of OD600 = 200 heat-killed *E. coli* and *M. luteus* (final OD600 = 100 for each), and frozen at –20°C 6, 12, and 24 hr post-infection. Total RNA was then extracted from pooled samples of five flies using TRIzol reagent per manufacturer's protocol and resuspended in MilliQ dH2O. Reverse transcription was performed using PrimeScript RT (Takara) with random hexamer and oligo dT primers. Quantitative PCR was performed on a LightCycler 480 (Roche) in 96-well plates using Applied Biosystems PowerUP SYBR Green Master Mix (Applied Biosystems). Data points represent pooled samples from three replicate experiments. Error bars represent one standard deviation from the mean. Statistical analyses were performed using one-way ANOVA with Holm–Sidak multiple test correction. Primers used in this study were:

Drs-F 5'-CGTGAGAACCTTTTCCAATATGAT-3'
Drs-R 5'-TCCCAGGACCACCAGCAT-3'
DptA-F: 5'-GCTGCGCAATCGCTTCTACT-3'
DptA-R: 5'-TGGTGGAGTGGGCTTCATG-3'
RPL32-F: 5'-GACGCTTCAAGGGACAGTATCTG-3'
RPL32-R: 5'-AAACGCGGTTCTGCATGAG-3'

## Ex vivo larval hemocyte phagocytosis assays

Ex vivo phagocytosis assays were performed using a mix of equal volumes of *E. coli* and *S. aureus* AlexaFluorTM488 BioParticlesTM (Invitrogen), following manufacturer's instructions; and see *Melcarne et al., 2019b*. Five L3 wandering larvae from our newly generated mutant lines, or carrying the *Hml-Gal4, UAS-GFP* hemocytes marker as a control, were bled into 150 µl of Schneider's insect medium (Sigma-Aldrich) containing 1 µM phenylthiourea (Sigma-Aldrich). The hemocyte suspension

was then transferred to 1.5 ml low binding tubes (Eppendorf, Sigma-Aldrich), and AlexaFluorTM488 bacteria BioParticlesTM were added. The samples were vortexed, incubated at room temperature for 2 hr to allow phagocytosis, and then placed on ice to stop the reaction. The fluorescence of extracellular particles was quenched by adding 0.4% trypan blue (Sigma-Aldrich) diluted to 1/3 concentration. Phagocytosis was quantified using a flow cytometer (BD Accuri C6) to measure the fraction of cells phagocytosing and their fluorescence intensity. Isogenic wild-type iso $w^{1118}$ larvae and *Hml-Gal4, UAS-GFP* larvae with or without bacterial particles were used to define the gates for hemocyte counting and the thresholds for phagocytosed particle emission. The phagocytic index was calculated as follows:

$$\text{Fraction of haemocytes phagocytosing(f)} = \frac{[\text{number of haemocytes in fluorescence positive gate}]}{[\text{total number of haemocytes}]}$$

$$\text{phagocyticindex}(PI) = [\text{Mean fluorescence intensity of haemocytes in fluorescence positive gate}] \times f$$

Finally, due to experimenter and experiment batch differences in total hemocytes collected and knock-on effects to calculating phagocytic index, phagocytic index was normalized with the wild-type as a reference set to 100% within experiment batch.

## Wounding experiment

A clean injury was performed with a needle sterilized with an EtOH and PBS wash. For imaging of the melanization reaction upon pricking, the thorax of 3- to 8-day-old flies was pricked using a sterile needle (diameter ~0.1 mm). Pictures were taken 24 hr post-pricking and categorized into normal, weak, or no melanization blackening seen at the injury site per *Dudzic et al., 2019*.

## Systemic infections and survival

Systemic infections were performed by pricking 3- to 5-day-old adult males in between the thoracic pleura with a 0.1-mm-thick needle dipped into a concentrated pellet of bacteria, yeast, or fungal spores. For natural infections with *A. fumigatus*, flies were anesthetized and then shaken on a sporulating plate of fungi for 30 s. For *B. bassiana*, flies were shaken for 30 s in a tube containing an excess of commercial spores sufficient to coat all flies uniformly, of which the flies remove the vast majority later through grooming. At least two replicate survival experiments were performed for each infection, though in specific cases some genotypes may have been included in only one experiment. 20–35 flies were included per vial when possible, kept on cornmeal fly media containing (per 1 l): 6.2 g agar, 58.8 g cornmeal, 58.8 g yeast, 60 ml grapefruit juice, 4.83 ml propionic acid, and 26.5 ml 100 g/l moldex in pure ethanol. Survivals were scored daily, and flies were flipped to new food vials 3 times per week. Due to challenges working with genotypes of different viability, we could not include all genotypes in all experimental replicates. Moreover, comparisons across genotypes required multiple levels of consideration. Given the complexity of genotype-vs-reference-by-pathogen comparisons in our study, needing to compare infections both to clean injury within genotype and to wild-type or *DITPM* across treatments, we chose to focus on major differences apparent in summary statistics, highlighting survival differences only when they were: (i) consistent across experimental replicates; (ii) of a consistent logic across comparable genotypes; for instance, compound mutants containing *DMel* (e.g. *DIMD*, *DMel*) should be as or more susceptible to infections as *DMel* alone if melanization is truly critical for defense; and (iii) of a mean lifespan difference ≥1.0 days after accounting for comparisons with unchallenged or clean-injury data. Total experiments (*N* exp) and total flies per genotype are reported to provide summary statistics per pathogen-by-genotype interaction. The mean lifespan reported in survival summary data has a maximum of 7 or 10 days depending on experimental conditions. Kaplan–Meier survival curves for all experiments are provided in the main text or supplementary information.

## Quantification of microbial load for growth kinetics

Three- to eight-day-old flies were infected with the indicated microbe and concentration per OD600 as described in *Supplementary file 1*, and allowed to recover. At the indicated time post-infection, flies were anesthetized using $CO_2$ and surface sterilized by washing them in 70% ethanol. Ethanol was removed, and then flies were homogenized using a Precellys bead beater at 6500 rpm for 30 s in LB broth (*Pr. rettgeri*), BHI (*S. aureus*), or YPG (*C. albicans*) with 500 µl as pools of five flies. These homogenates were serially diluted and 100 µl was plated on LB, BHI, or YPG agar. Plates were incubated overnight, and colony-forming units were counted using an Interscience Scan 500 plate scanning colony counter and validated independently by manual counts for a subset of plates to ensure accuracy. Statistical differences were tested using Brown–Forsythe and Welch ANOVA tests with Dunnett's multiple test correction, which consider unequal variance.

## Acknowledgements

We thank Mélanie Blokesch (EPFL), Jan-Willem Veening (Unil), Vivek Thacker (Heidelberg University), Carla Saleh (Pasteur Institut), and Ronald P van Rij (Radboud University Medical Center) for key reagents. We thank Luis Teixeira for iso Drosdel wild-type, *Rel*, and *spz* flies. We thank Elodie Koenig, Prince Kumar Sah, and Hannah Westlake for experimental help and Hannah Westlake for editing. The AMP, Bom, and single and quadruple module mutants were deposited at the Vienna *Drosophila* Research Center. This project was supported by the SNSF grant 310030_215073 awarded to BL, and Wellcome Trust grant 227559/Z/23/Z awarded to MAH.

## Additional information

### Competing interests

Bruno Lemaitre: Reviewing editor, eLife. The other authors declare that no competing interests exist.

### Funding

| Funder | Grant reference number | Author |
| --- | --- | --- |
| Schweizerischer Nationalfonds zur Förderung der Wissenschaftlichen Forschung | 310030_215073 | Bruno Lemaitre |
| Wellcome Trust | 10.35802/227559 | Mark Austin Hanson |

The funders had no role in study design, data collection, and interpretation, or the decision to submit the work for publication. For the purpose of Open Access, the authors have applied a CC BY public copyright license to any Author Accepted Manuscript version arising from this submission.

### Author contributions

Faustine Ryckebusch, Conceptualization, Resources, Data curation, Investigation, Methodology, Writing – original draft; Yao Tian, Conceptualization, Investigation, Methodology; Mylene Rapin, Data curation, Investigation; Fanny Schüpfer, Investigation; Mark Austin Hanson, Conceptualization, Resources, Data curation, Formal analysis, Supervision, Validation, Visualization, Project administration, Writing – review and editing; Bruno Lemaitre, Conceptualization, Resources, Supervision, Funding acquisition, Methodology, Writing – original draft, Project administration

### Author ORCIDs

Faustine Ryckebusch ⓘ https://orcid.org/0000-0002-9626-884X
Mark Austin Hanson ⓘ https://orcid.org/0000-0002-6125-3672
Bruno Lemaitre ⓘ https://orcid.org/0000-0001-7970-1667

Reviewer #1 (Public review): https://doi.org/10.7554/eLife.107030.3.sa1

Reviewer #2 (Public review): https://doi.org/10.7554/eLife.107030.3.sa2
Author response https://doi.org/10.7554/eLife.107030.3.sa3

## Additional files

### Supplementary files

Supplementary file 1. Microbe characteristics and growth conditions.

Supplementary file 2. Survival curves for data summarized in *Figure 3*.

Supplementary file 3. Survival analysis outputs for data summarized in *Figure 3*.

Supplementary file 4. Supplemental discussion of fly genetics and considerations for the use of genotypes described in this study.

MDAR checklist

### Data availability

All data analyzed in this manuscript are provided in *Supplementary file 1–4*. The *Drosophila* reagents generated in this study have been deposited in the Vienna *Drosophila* Resource Center as the DrosDel Immunity Panel: https://shop.vbc.ac.at/vdrc_store/vdrc-fly-stocks/drosdel-immunity-panel.html.

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
