## [Editor Report · eLife Assessment]

This work provides one of the first **important** attempts to look at *Drosophila* immune responses against bacterial, viral, and fungal pathogens in a way that combines the roles of four major arms in immunity (Imd signaling, Toll signaling, phagocytosis, and melanization) rather than studying them separately. The findings are **compelling** and the tools provided can be used as they are, or built upon, in various contexts.

---

## [Referee Report · Reviewer #1 (Public review)]

Summary:

The innate immune system serves as the first line of defense against invading pathogens. Four major immune-specific modules-the Toll pathway, the Imd pathway, melanization, and phagocytosis-play critical roles in orchestrating the immune response. Traditionally, most studies have focused on the function of individual modules in isolation. However, in recent years, it has become increasingly evident that effective immune defense requires intricate interactions among these pathways.

Despite this growing recognition, the precise roles, timing, and interconnections of these immune modules remain poorly understood. Moreover, addressing these questions represents a major scientific undertaking.

Strengths:

In this manuscript, Ryckebusch et al. systematically evaluate both the individual and combined contributions of these four immune modules to host defense against a range of pathogens. Their findings significantly enhance our understanding of the layered architecture of innate immunity.

---

## [Referee Report · Reviewer #2 (Public review)]

Summary:

In this work, the authors take a holistic view at the *Drosophila* immunity by selecting four major components of fly immunity often studied separately (Toll signaling, Imd signaling, phagocytosis and melanization), and studying their combinatory effects on the efficiency of the immune response. They achieve this by using fly lines mutant for one of these components, or modules, as well as for a combination of them, and testing the survival of these flies upon infection with a plethora of pathogens (bacterial, viral and fungal).

Strengths:

It is clear that this manuscript has required a large amount of hands-on work, considering the number of pathogens, mutations and timepoints tested. In my opinion, this work is a very welcome addition to the literature on fly immune responses, which obviously do not occur one type of a response at a time, but in parallel, subsequently and/or are interconnected. I find that the major strength of this work is the overall concept, which is made possible by the mutations designed to target the specific immune function of each module, without effects on other functions. I believe that the combinatory mutants will be of use for the fly community and enable further studies of interplay of these components of immune response in various settings.

To control for the effects arising from the genetic variation other than the intended mutations, the mutants have been backcrossed into a widely used, isogenized *Drosophila* strain called w1118. Therefore, the differences accounted for by the genotype are controlled.

I also appreciate that the authors have investigated the two possible ways of dealing with an infection: tolerance and resistance, and how the modules play into those.

Weaknesses:

While controlling for the background effects is vital, the w1118 background is problematic (an issue not limited to this manuscript) because of the wide effects of the white mutation on several phenotypes (also other than eye color/eyesight). It is a possibility that the mutation influences the functionality of the immune response components. I acknowledge that it is not reasonable to ask for data in different backgrounds better representing a "wild type" fly, but I think this matter should be brought up and discussed.

The whole study has been conducted on male flies. Immune responses show quite extensive sex-specific variation across a variety of species studied, also in the fly. But the reasons for this variation are not fully understood. Therefore, I suggest that the authors would conduct a subset of experiments on female flies to see if the findings apply to both sexes, especially the infection-specificity of the module combinations.

Comments on the revised manuscript:

I appreciate the author's responses to the points I raised and the additional work they have conducted. The authors have now discussed the possible background effect and added an experiment on female flies showing that the module function is applicable to both sexes.

---

## [Author Response]

The following is the authors’ response to the original reviews.

**Reviewer #1 (Public review):**
Summary:The innate immune system serves as the first line of defense against invading pathogens. Four major immune-specific modules - the Toll pathway, the Imd pathway, melanization, and phagocytosis- play critical roles in orchestrating the immune response. Traditionally, most studies have focused on the function of individual modules in isolation. However, in recent years, it has become increasingly evident that effective immune defense requires intricate interactions among these pathways.Despite this growing recognition, the precise roles, timing, and interconnections of these immune modules remain poorly understood. Moreover, addressing these questions represents a major scientific undertaking.Strengths:In this manuscript, Ryckebusch et al. systematically evaluate both the individual and combined contributions of these four immune modules to host defense against a range of pathogens. Their findings significantly enhance our understanding of the layered architecture of innate immunity.

We thank the reviewer for their kind assessment.

Weaknesses:While I have no critical concerns regarding the study, I do have several suggestions to offer that may help further strengthen the manuscript. These include:(1) Have the authors validated the efficiency of the mutants used in this study? It would be helpful to include supporting data or references confirming that the mutations effectively disrupted the intended immune pathways.

We have done so in Figure 1.

(2) Given the extensive use of double, triple, and quadruple mutants, a more detailed description of the mutant construction process is warranted.

We now provide a supplement (File S1) that details the successive genetic crosses and recombinations that were required to generate these compound fly stocks carrying multiple mutations. We also provide some information regarding rapid screening of stocks for phenotypes. Of note some of these fly stocks have been deposited at VDRC as they will be useful to fly community to assess immune modules in a controlled background, and complete stock information will be tied to these stocks there.

**Reviewer #2 (Public review):**
Summary:In this work, the authors take a holistic view of *Drosophila* immunity by selecting four major components of fly immunity often studied separately (Toll signaling, Imd signaling, phagocytosis, and melanization), and studying their combinatory effects on the efficiency of the immune response. They achieve this by using fly lines mutant for one of these components, or modules, as well as for a combination of them, and testing the survival of these flies upon infection with a plethora of pathogens (bacterial, viral, and fungal).Strengths:It is clear that this manuscript has required a large amount of hands-on work, considering the number of pathogens, mutations, and timepoints tested. In my opinion, this work is a very welcome addition to the literature on fly immune responses, which obviously do not occur in one type of response at a time, but in parallel, subsequently, and/or are interconnected. I find that the major strength of this work is the overall concept, which is made possible by the mutations designed to target the specific immune function of each module (at least seemingly) without major effects on other functions. I believe that the combinatory mutants will be of use for the fly community and enable further studies of the interplay of these components of immune response in various settings.To control for the effects arising from the genetic variation other than the intended mutations, the mutants have been backcrossed into a widely used, isogenized *Drosophila* strain called w1118. Therefore, the differences accounted for by the genotype are controlled.I also appreciate that the authors have investigated the two possible ways of dealing with an infection: tolerance and resistance, and how the modules play into those.

We thank the reviewer for their kind assessment.

Weaknesses:While controlling for the background effects is vital, the w1118 background is problematic (an issue not limited to this manuscript) because of the wide effects of the white mutation on several phenotypes (also other than eye color/eyesight). It is a possibility that the mutation influences the functionality of the immune response components, for example, via effects of the faulty tryptophan handling on the metabolism of the animal.I acknowledge that it is not reasonable to ask for data in different backgrounds better representing a "wild type" fly (however, that is defined is another question), but I think this matter should be brought up and discussed.

We agree with the reviewer and have included caveats on the different genetic effects brought about the combinatory mutant approach including differences in white gene status, insertion of GFP or DsRed markers, and nature of genetic mutations (Line 142-on).

“Of note, the strains used in this study differ in their presence/absence of the white^+^ gene, present in the PPO1^∆^, NimC1^1^ and eater^1^ mutations. In addition to its well established function in eye pigmentation, the white gene can also impact host neurology and intestinal stem cell proliferation (Ferreiro et al., 2017; Sasaki et al., 2021). We did not observe any obvious correlations between white^+^ gene status and susceptibilities in this study. Moreover, in a previous study looking at the cumulative effects of AMP mutations on lifespan, white gene status and fluorescent markers did not readily explain differences in longevity (Hanson and Lemaitre, 2023). We therefore believe that the extreme immune susceptibility we have created through deficiencies for pathways regulating hundreds of genes, or major immune modules, overwhelms the potential effects of white^+^ and other transgenic markers. For additional information on which stocks bear which markers, see discussion in Supplementary file 1.”

Of interest, we were highly conscious of this concern in working with combinatory AMP mutants which differed in white, GFP, and DsRed copies. However, even over the many weeks of snowballing effects on microbiota community composition and structure, we found no trends tied strictly to white+ or to other genetic insertions on lifespan (Hanson and Lemaitre, 2023; DMM).

The whole study has been conducted on male flies. Immune responses show quite extensive sex-specific variation across a variety of species studied, also in the fly. But the reasons for this variation are not fully understood. Therefore, I suggest that the authors conduct a subset of experiments on female flies to see if the findings apply to both sexes, especially the infection-specificity of the module combinations.

We thank the reviewer for this suggestion. We have performed the requested experiments, and include female survival trends in Figure 4supp1. We have added the following text to the main manuscript (Line 554):

“All survival experiments to this point were done with males. We therefore assessed key survival trends for these infections in females to learn whether the dynamics we observed were consistent across sexes (Figure 4supp1). For all three pathogens (Pr rettgeri, Sa aureus, C. albicans) the rank order of susceptibility was broadly similar between males and females, with higher rates of mortality in females overall. Thus, we found no marked sex-bygenotype interaction. Interestingly, the greater susceptibility of females in our hands is true even for ∆ITPM flies, although there are only a few surviving flies on which we can base these conclusions. However, these data may suggest the sexual dimorphism in defense against infection that we see against these pathogens is due to factors independent of the immune modules we disrupted.”

It is worth noting that male-female sex dichotomies in infection are inconsistent across the literature, with strong lab-specific effects (Belmonte et al., 2020 and personal observation). In our lab setting, we consistently see female mortality higher than males when compared, independent of pathogen and mutant background. We have not seen notable interaction terms of sex and genotype for most immune deficient mutants. It is quite interesting to have done these experiments with ITPM, however, which reveals that there is at least a trend suggesting this dichotomy is independent of the four immune modules we deleted. Still, our infection conditions kill most males, and so it would be good to replicate this sex-specific ∆ITPM result in a dedicated study with doses chosen to improve the resolution of male-female differences. For now, we prefer to use conservative language and avoid overinterpreting this trend, but do feel it merits mentioning.

**Recommendations for the authors:**

Comment on statistical requests

Both reviewers requested further clarity on the statistical analyses supplemental to Figure 3. We haved address these comments as follows.

First, we now provide an additional supplementary .zip file containing summary statistics for all survival data in Figure 3 (Supplementary File 3). We have additionally added this text to line 226 to make this data treatment more clear:

…” we chose to focus on major differences apparent in summary statistics,Highlighting”…

And we highlight that all survival data are also provided as Kaplan-Meier survival curves in the main or supplementary figures in Line 233:

“Kaplan-Meier survival curves for all experiments are provided in the main text or supplementary information”.

Second, as outlined in the main text, we were unable to sample across all pathogenby-genotype interactions systematically, and this unfortunately obfuscates robust statistical modelling. We addressed the challenge of finding meaningful statistical differences by focusing on trends only if they were (i) consistent across experimental replicates, (ii) of a consistent logic across comparable genotypes, ensuring random inter-experimental noise was not unduly shaping interpretations, and (iii) of a mean lifespan difference ≥1.0 days compared to wild-type, and compared to relevant unchallenged or clean-injury controls. This last choice was especially important because not all experimental replicates included all genotypes due to challenges of animal husbandry and coordination among multiple researchers over five years of data collection. As a result, our initial analyses using a cox mixed-effects model found it to be rather useless, being insensitive to important experiment batch effects visible to the eye because statistically-affected genotypes were not present in all experiments.

We therefore ensured that behaviour relative to controls within* experiments was consistent, rather than the comparison of genotypes to controls across the sum of experiments with a post-hoc treatment attempting to apportion variance to experiment batch (but unable to do so for some genotypes and some batches). Due to differeces in baseline health and the dynamics explained by studies like Duneau et al. 2017; eLife, there is an expected unequal variance of genotype*pathogen interactions across experiment batches. Unfortunately, this unequal variance, coupled with incomplete sampling across experiment batches, means “highly significant” differences can emerge that don’t hold up to scrutiny of comparisons to controls taken only from within an experiment batch. Thus, we chose to forego a cox mixed effect model approach entirely. Instead, our highly conservative approach, focusing on only very large effects with a mean lifespan difference ≥1.0 days, mitigates these issues. We have taken great care to ensure that any results we highlight stand up to inter-experiment batch effects. We would further draw the reviewers’ attention to our response to Reviewer 2 relating to Figure 3, which emphasizes the level of conservativism that we are applying.

At the end of the Discussion, we have added the following sentence to emphasize these limitations:

“…a combinatorial mutation approach to deciphering immune function can be extended even to the broad level of whole immune modules. Of note, we were unable to systematically sample all genotype-bypathogen interactions equally. We have therefore been highly conservative in our reporting of major effects. There are likely many important interactions” not discussed in our study. Future investigations may highlight important biology that is apparent in our data, but which we may not have mentioned here. To this end, we have deposited our isogenic immunity fly stocks in the Vienna *Drosophila* Resource Centre to facilitate their use. Beyond immunity, our tools can also be of use to study various questions at the cutting edge of aging, memory, neurodegeneration, cancer, and more, where immune genes are repeatedly implicated. We hope that this set of lines will be useful to the community to better characterize the *Drosophila* host defense.”

We recognise this response may not fully satisfy the reviewers’ requests. While use of summary statistics is simple, our rules for highlighting interactions of importance are defined, readily understood and interpreted, and draw attention to key trends in that are backed by a solid understanding of the data and its limitations. We have taken this approach out of a responsibility to avoid making spurious assertions that stem from underpowered statistical models rather than from the biology itself.

**Reviewer #1 (Recommendations for the authors):**
(1) Lines 1092-1093 - Please double-check the labeling of the panels in Figure 2. It appears that panels A and C correspond to single-module mutants, whereas panels B and D refer to compound-module mutants.

We have modified Figure 2 and Figure 2supp1 labelling. We also realise there was an error in the column titling that contributed to the confusion. We hope the new layout is clear, and thank the reviewers for noting this issue.

(2) Lines 347-377 - Figure 2D is not cited in the text.

We now cite Fig2D in Line 356.

(3) P values should be indicated in Figure 2 and Figure 3 for all relevant comparisons. Additionally, "ns" (not significant) should be added in Figure 5A-B.

We make the effort to show key uninfected survival trends in Figure 2, and list the total flies (n_flies) in Fig3 to provide the reader with the underlying confidence in the trends observed. We focus on differences of mean lifespan of at least 1 day, and which are consistent in direction across combinatory mutations. We have avoided the multiple comparisons of cox proportional hazard survival analyses throughout this study because they are overly sensitive for our purposes, as we have previously when systematically comparing many genotypes to each other (see Hanson and Lemaitre, 2023; DMM).

(4) Minor points: Hml-Gal4, UAS-GFP should be italic; Line 192-- "uL" and "uM"; Line 596: P>.05.

We have made these changes. We’re unsure what the comment regarding P>.05 referred to, but have removed spaces and made it non-italics.

**Reviewer #2 (Recommendations for the authors):**
Statistical analyses and their outcomes are clearly indicated only for the data in Figure 1 and Figure 5 and in the supplement for Figure 1, while they are not reported/not easily accessible for other data. For the main figures, statistics should be indicated in the figure for an easier assessment of the data. In case of multiple comparisons potentially crowding the plots too much, statistics may be in a supplementary file/table.

See response above.

In case of the hemocytes, besides phagocytosis, I would think that ROS generation via the DUOX/NOX system is also an integral part of the immune response against pathogens, and that has not been included here. That might be an interesting addition for future experiments. As the NimC1, eater double mutant flies are said to have fewer hemocytes, it is possible that this function of the hemocytes is affected as well. This could be commented on in the text.

The reviewer raises a good point. The role of DUOX and NOX in ROS responses is not assessed in our study. To our knowledge, DUOX and NOX participate primarily in the wound repair response, or in epithelial renewal at damage sites or in the gut. In our study on systemic immunity, we did not assess the role of clotting, the precise function of ROS, and we have missed other host defense or stress response mechanisms as well (e.g. constitutively-expressed AMP-like genes, TEPs, JAK-STAT) that likely play a role in the systemic immune defense. Considering the lethality caused by Nox and Duox mutation, there would be inherent genetic difficulties to recombine these as multiple mutations. Unfortunately, this makes it difficult to include these processes in our analysis in a systematic manner. We are already happy to have generated fly lines lacking four immune modules simultaneously, even if they are not fully immune deficient. We have mentioned this point in the discussion (Line 613-on).

Of note, the NimC1, eater double mutants actually have decreased hemocyte counts at the adult stage (Melcarne et al,. 2019). Thus NimC1, eater double mutants are not impaired only in phagocytosis, but the overall cellular response. We make a point to outline this in Line 225-257, and 607.

I think it could be mentioned that the melanization response at larval stage (against parasitoids) functions differently from the melanization described here (requiring hemocyte differentiation and PPO3).

A good point. We have added this mention in Line 97:

“In addition, a third PPO gene (PPO3) is specifically expressed by lamellocytes, specialized hemocytes that differentiate in larvae responding to and enveloping invading parasites (Dudzic et al., 2015)”.

Overall, the clarity of the figures and figure legends could be worked on to make them a bit easier to follow. Below are some of my suggestions:(1) In Figure 2, adding headings to parts C & D (similarly to A & B) would make it easier to follow what is happening in the figure at a glance. Also, it is rather difficult to visually follow which strain is which in the plots. I'd suggest adding the key/legend for single mutants below 2A & B, and the key for the double mutants below C & D. If a mutant is present in A & B and in C & D, it could be included in both keys. I also think that it would be intuitive to present the single mutants by dashed lines and double mutants by continuous lines (or vice versa), so that one would easily distinguish between them. Of note, the figure legend says that A & B are single mutants, but for example in B there are also some double mutants (?).

We have modified Figure 2 and Figure 2supp1 labelling. We also realise there was an error in the column titling that contributed to the confusion. We hope the new layout is clear, and thank the reviewers for noting this issue.

(2) In Figure 3, it looks like ΔMel is almost identical to controls in the clean injury survival, but in Figure 2C, it is clearly doing worse. I might be missing something here, but would like the authors to clarify the matter. Also, the meaning of the numbers in the heat map could be explained in the figure legend and/or added to the figure (color key).

The reviewer is correct. We thank the reviewer for this astute observation. Inadvertently, we used an old version of the Figure 2 preparation where only a subset of experiments was entered in the Prism data file rather than the total data used to inform Figure 3. This issue affected all genotypes.

We have reviewed the data in Figure 2, Figure 2supp1, and Figure 3, and updated these figures accordingly to ensure they represent the full survival data. We have also incorporated new experiments into the sum data related to male-female differences and to fill gaps in the data from the 1^st^ submission. We will also note due to the nature of 1^st^ decimal rounding that the difference between WT and ΔMel appears slightly underrepresented: the true difference (over the 7-day lifespan) is 0.37. We’ve provided a version of this figure rounded to 2 decimal places below, but prefer the simpler 1 decimal place in the main text for readability. The updated Figure 2 shows the full data in Figure 3 accurately.

We will also take this opportunity to highlight how conservative our ≥1.0 days difference approach is. Breaking down survival curve patterns in Figure 2 relative to mean differences in Figure 3, for clean injury, approximately ~75% of ΔMel flies survive to day 7 with mortality mostly taking place between days 3-7. The result is a mean lifespan of 6.37 days. On a survival curve, this difference appears quite strong, but in our mean lifespan table the difference is rather muted (WT vs. ΔMel difference = 0.37 days). Thus, differences of ≥1.0 days reflect very strong trends in survival data that are near-guaranteed to be independent of experimental noise. While we note issues that prevented us from a fully systematic sampling for all experiments, we are confident that the ≥1.0 day differences we highlight, using the rules explained in the main text, are robust. While this approach could be seen as overly conservative, it is our preference in this initial study, containing combinations of 25 treatments and 14 genotypes, to be highly conservative. Future studies may investigate other strong differences we have not highlighted, and the data we provide here can help generate expectations and guide those studies.

**Author response image 1. sa3fig1:** Figure 3 with 2 decimals places of rounding for mean lifespans. The 7-day clean injury mean lifespan of WT is 6.74 days, and of ΔMel is 6.37 days. Due to rounding, in the 1 decimal Figure 3 this difference appears as if it is only 0.3 days, but it closer to 0.4 days. Regardless, this level of difference, which appears rather clearly in a survival curve, is well below the level of difference we have chosen to highlight in our study.

(1) Figure 4: I find it very tedious to compare CFUs among different mutants from the plots. As the idea is to compare bacterial loads among the mutants at different timepoints, it would be easier to compare them if the data were shown within a timepoint (CFUs of each mutant at 2h, at 6h, and so on). This is also how the results are written in the text (within a time point). Would it also be clearer if the CFU plots were named, for example: " A', B', and C'"?

We appreciate this note. We feel both representations have merits and pitfalls, but prefer our original design showing the progression of bacterial growth within genotype first. However, we have added dotted lines representing the wild-type bacterial loads at 2hpi, 12hpi, and 24hpi to assist the reader in making acrossgenotype comparisons at key time points. Like this, the reader can see if the error bars (StDev) overlap the mean of the wild-type, and so make more intuitive judgements about whether these differences are meaningful.

(2) Figure 2D is not referred to in the text.

We now cite Fig2D in Line 356.